# LATENCY-AWARE CONTEXTUAL BANDIT: APPLICATION TO CRYO-EM DATA COLLECTION

## ABSTRACT

We introduce a latency-aware contextual bandit framework that generalizes the standard contextual bandit problem, where the learner adaptively selects arms and switches decision sets under action delays. In this setting, the learner observes the context and may select multiple arms from a decision set, with the total time determined by the selected subset. The problem can be framed as a special case of semi-Markov decision processes (SMDPs), where contexts and latencies are drawn from an unknown distribution. Leveraging the Bellman optimality equation, we design the contextual online arm filtering (COAF) algorithm, which balances exploration, exploitation, and action latency to minimize regret relative to the optimal average-reward policy. We analyze the algorithm and show that its regret upper bounds match established results in the contextual bandit literature. In numerical experiments on a movie recommendation dataset and cryogenic electron microscopy (cryo-EM) data, we demonstrate that our approach efficiently maximizes cumulative reward over time.

## 1 INTRODUCTION

The contextual bandit framework models sequential decision-making under uncertainty: the learner observes context, selects an action, and receives feedback only for that action (Lattimore & Szepesvári, 2020). This framework is widely used in domains requiring personalization, experimentation, or optimization under uncertainty, including recommender systems, healthcare, education, finance, and energy management (Li et al., 2010; Tewari & Murphy, 2017; Lan & Baraniuk, 2016; Soemers et al., 2018; Chen et al., 2020). Standard formulations do not account for the latency of acquiring information or executing actions. In practice, obtaining contexts—such as assay results, medical records, or experimental measurements—often involves non-negligible delays. Similar challenges arise in scientific automation, where experimental decisions must trade off information gain against time constraints. Examples include high-throughput drug discovery, automated materials science, and astronomy (Blay et al., 2020; Pyzer-Knapp et al., 2022; Adler et al., 2020). A prominent instance is cryo-electron microscopy (cryo-EM) (Li et al., 2023), where limited and costly microscope time must be efficiently allocated to the most informative imaging targets.

To address this limitation, we extend the contextual bandit model to incorporate latency-aware decision-making. At each round, the learner selects multiple arms from a decision set and receives their rewards, with a total time cost determined by the chosen subset. Maximizing cumulative reward in this setup requires balancing the trade-off between *exploration* (gathering information from new actions) and *exploitation* (selecting actions with high expected rewards), while also implementing an effective strategy for arm selection under latency. This problem can be framed as a special case of SMDPs where the reward function, sojourn time distribution, and transition probabilities are unknown, making it a reinforcement learning task. We adopt the framework of undiscounted reinforcement learning (Auer et al., 2008) under the average reward criterion. In particular, we make the following contributions:

- We analyze the latency-aware contextual bandit problem and derive the Bellman optimality equation to characterize the optimal policy. We show that the maximum average reward can be obtained by finding the root of a function with noisy measurements.

- Building on the Bellman optimality equation, we leverage stochastic approximation and the upper confidence bound (UCB) method to design the COAF algorithm, which efficiently selects arms

and switching decision sets under action latency. We establish that COAF achieves sublinear regret and validate its performance through numerical experiments on a movie recommendation dataset and cryo-EM data collection. The results demonstrate the effectiveness of COAF in time-sensitive decision-making tasks.

## 2 RELATED WORKS

In sequential decision-making under uncertainty, maximizing reward requires balancing exploration and exploitation. The multi-armed bandit (MAB) problem formalizes this trade-off: a learner repeatedly selects arms with unknown reward distributions and aims to minimize regret, defined as the difference between the cumulative reward of an online algorithm and that of the optimal arm (Robbins, 1952; Lai & Robbins, 1985). Classical algorithms with strong theoretical guarantees include the UCB family, which selects arms according to optimistic estimates of their mean rewards. Thompson sampling, one of the earliest MAB solutions, is a randomized Bayesian algorithm that nonetheless addresses the fundamentally frequentist problem of regret minimization (Thompson, 1933; Agrawal & Goyal, 2012; Kaufmann et al., 2012). Other widely studied policies include the Gittins index (Gittins et al., 2011; Lattimore, 2016) and minimum empirical divergence (Honda & Takemura, 2010).

The contextual bandit problem generalizes the MAB by allowing the learner to make decisions based on observed contexts. This framework naturally integrates statistical learning and function approximation into sequential decision-making. Contextual bandit algorithms can be broadly categorized into two types. *Realizability-based approaches* assume that rewards follow a known parametric family, enabling efficient algorithms with strong theoretical guarantees. Representative examples include LinUCB and linear Thompson sampling for linear models (Chu et al., 2011; Agrawal & Goyal, 2013), and GLM-UCB, GLM-TS, and GLOC for generalized linear models (Filippi et al., 2010; Abeille & Lazaric, 2017; Jun et al., 2017). In contrast, *general-purpose approaches* make weaker assumptions, accommodating broader function classes. They often rely on regression oracles, with regret bounds expressed in terms of sample complexity measures such as VC-dimension, eluder dimension, or the performance of a square-loss minimizing oracle (Langford & Zhang, 2007; Beygelzimer et al., 2011; Russo & Van Roy, 2013; Foster & Rakhlin, 2020). Empirically, realizability-based methods outperform general-purpose approaches when the reward model is well-specified, while the latter offer greater flexibility under unknown or complex reward structures (Bietti et al., 2021).

Our problem formulation allows the learner to select multiple arms from a decision set. This setup was first introduced by Anantharam et al. (1987) and is widely studied in combinatorial bandits. The reward function can be linear with respect to the individual arm rewards (Cesa-Bianchi & Lugosi, 2012) or nonlinear, capturing interactions and combinatorial constraints (Chen et al., 2013; Kveton et al., 2014; Chen et al., 2016). Contextual combinatorial bandits focus on learning the combinatorial reward structure under context, where decision sets arrive sequentially and selecting a combination of arms incurs a fixed time cost (Qin et al., 2014). This model does not explicitly account for action latency. We adopt the semi-bandit feedback model (Kveton et al., 2015), where the learner receives granular feedback in the form of individual rewards for each selected arm.

The idea of switching decision sets in our problem is inspired by the mortal MAB (Chakrabarti et al., 2008), where the learner can request new decision sets, and the lifetime of each set (i.e., the number of available arms) follows a geometric distribution. Similarly, the sleeping experts problem (Kanade et al., 2009; Kleinberg et al., 2010) considers a dynamic arm set, where arms are activated either stochastically or by an adversary. In this setup, the learner passively reacts to the changes of arm sets. In contrast, as in mortal MAB, our formulation allows the learner to actively control the dynamics by switching to new decision sets, potentially accessing better arms. Our problem integrates dynamic control of action space into contextual decision-making, combining elements of mortal MAB and contextual combinatorial bandits. This adds complexity, as the learner must balance both expected reward and the time required for each decision.

## 3 PROBLEM SETTINGS

This section presents the formal problem formulation for the latency-aware contextual bandit and discusses connections to existing works. As a natural application, cryo-EM data collection is introduced, where microscope operations induce inherent latencies.

### 3.1 LATENCY-AWARE CONTEXTUAL BANDITS

We consider a latency-aware contextual bandit problem. At each round $j = 1, 2, \dots$:

- The learner observes: (i) the arm feature vectors $\mathbb{X}_j = \{\mathbf{x}_{j,1}, \dots, \mathbf{x}_{j,n_j}\} \subset \mathbb{R}^d$; (ii) the action space $\mathbb{A}_j \subseteq 2^{[n_j]}$ containing subsets of arms; and (iii) a latency function $l_j : \mathbb{A}_j \to \mathbb{R}_{\geq 0}$.
- The learner selects a subset of arms $A_j \in \mathbb{A}_j$ and observes semi-bandit feedback: for each $i \in A_j$, the reward $y_{j,i}$ is revealed, while the rewards of unchosen arms remain unknown.
- The learner receives total reward $r_j(A_j) = \sum_{i \in A_j} y_{j,i}$[1], and the time spent is $l_j(A_j)$.

Several elements of the setup are stochastic, including $\mathbb{X}_j$ (and its size $n_j$), $\mathbb{A}_j$, and $l_j$. We assume that the sequence $\{(\mathbb{X}_j, \mathbb{A}_j, l_j)\}_{j=1}^{\infty}$ is IID, with each $(\mathbb{X}_j, \mathbb{A}_j, l_j)$ drawn from an unknown distribution $P_{\text{env}}$. Arbitrary dependencies among $\mathbb{X}_j$, $\mathbb{A}_j$, and $l_j$ within a round $j \in \mathbb{N}$ are allowed. Each selected arm $i$ yields a random reward $y_{j,i} = \psi_*(\mathbf{x}_{j,i}) + \epsilon_{j,i}$, where $\psi_* : \mathbb{R}^d \to [-1, 1]$ is the bounded mean reward function unknown to the learner, and noise $\epsilon_{j,i}$ is a zero-mean random variable. We further impose the following assumptions.

**Assumption 1** (Boundedness). *There exist $n_{\max}, l_{\min}, l_{\max} > 0$ such that for all rounds $j \in \mathbb{N}$: (i) the number of arms $n_j \leq n_{\max}$; (ii) the action time $l_j(A_j) \in [l_{\min}, l_{\max}]$ for all $A_j \in \mathbb{A}_j$.*

**Assumption 2** (Realizability (Foster et al., 2018)). *The learner is given a regressor class $\mathcal{F}$ that contains the bounded mean reward function $\psi_*$, i.e., $\psi_* \in \mathcal{F}$.*

**Remark 1.** *The term $l_j(\emptyset)$ can be interpreted as the time spent to acquire contextual information, and the condition $l_j(A_j) \geq l_{\min} > 0$ for all $A_j \in \mathbb{A}_j$ ensures temporal progress at each round $j$. Under Assumption 2, it suffices to learn the mean reward function within the regressor class $\mathcal{F}$.*

The problem is formally specified by $\mathcal{M} = (P_{\text{env}}, \psi_*)$, and the objective is to maximize cumulative reward without prior knowledge of $\mathcal{M}$. Beyond the standard exploration–exploitation trade-off in MAB problems, the learner must balance exploiting the current decision set, where good arms may be exhausted, with switching to new sets, taking action latency into account.

### 3.2 RELATIONSHIP WITH EXISTING WORKS

The problem studied in this paper generalizes the following existing bandit setups.

**Stochastic contextual bandits (Lattimore & Szepesvári, 2020):** In this setup, the learner observes the context of arm features $\mathbb{X}_j$ and selects a single arm at each round $j$. This corresponds to a special case of our problem, where $\mathbb{A}_j = \{\{1\}, \dots, \{n_j\}\}$ and the action time $l_j(A_j) = 1$ for all $A_j \in \mathbb{A}_j$.

**Contextual combinatorial semi-bandits (Qin et al., 2014):** This formulation, like our problem, allows the learner to select subsets of arms $A_j \in \mathbb{A}_j \subseteq 2^{[n_j]}$ at each round $j$, but does not explicitly model action latency, i.e., $l_j(A_j) = 1$ for all $A_j \in \mathbb{A}_j$.

**Mortal MAB (Chakrabarti et al., 2008):** In this setup, all arms in a decision set have identical rewards, $y_{j,i} = y_j$ for all $i \in [n_j]$, and the sequence $\{y_j\}_{j=1}^{\infty}$ is IID with $y_j \sim P_y$. The number of arms $n_j$, drawn from a geometric distribution with parameter $p$, represents the lifetime of the decision set. Our problem generalizes this setup by incorporating contextual information and allowing heterogeneous rewards across arms. The mortal MAB is a planning problem, as both $P_y$ and $p$ are assumed known, whereas our setting is a learning problem with unknown $P_{\text{env}}$ and $\psi_*$.

### 3.3 CRYO-EM DATA COLLECTION

Single-particle cryo-EM is a structural biology technique for determining near-atomic resolution 3D structures of biomolecules. A purified sample is applied to a thin, electron-transparent grid and rapidly frozen in vitreous ice. Imaging produces 2D projections of particles by passing an electron beam through the sample. A typical data collection workflow is illustrated in Fig. 1. The grid contains multiple *squares*, each with several *holes* where biomolecules are preserved in thin ice.

---

[1]While $r_j(A_j)$ can be generalized to be non-linear under monotonicity and Lipschitz assumptions, as in Qin et al. (2014), we focus on the linear case, as handling arm interactions is beyond the scope of this work.

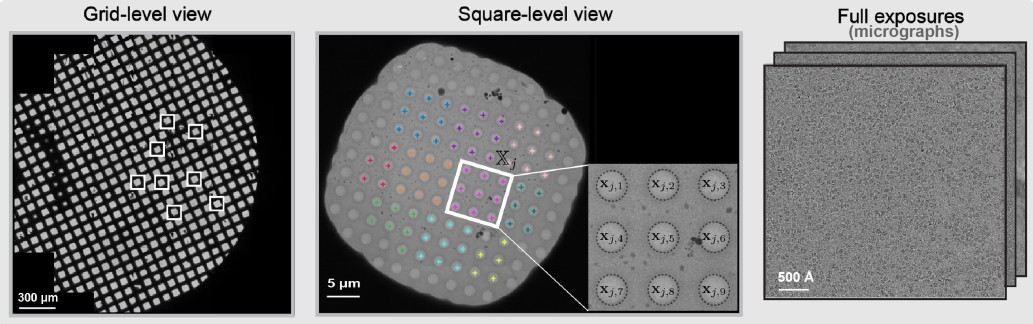

Figure 1: Cryo-EM data collection at multiple magnifications: (i) *grid-level* shows the entire grid at low magnification, (ii) *square-level* captures individual squares at medium magnification to assess ice quality within holes, and (iii) *full exposures* are high-magnification images of selected holes.

Grid-level and square-level views are used to navigate and select holes for *full exposures*. These high-magnification exposures, taken with high electron doses, are used for 3D reconstruction of biomolecules. Radiation irreversibly damages the sample, so each region can be imaged only once.

Cryo-EM data collection is inherently a bandit problem with partial feedback: selecting a set of holes reveals the data quality only for the chosen holes, while unselected holes remain unknown. Our latency-aware formulation captures the time required for exposures, refocusing, and stage movements. Neighboring holes can often be imaged via fast beam shifts, but larger movements require physically moving the stage, which is slower and necessitates additional refocusing. To capture this, holes in a square are divided into patches (colored in Fig. 1), each forming a decision set of $n_j$ arms, with contexts $\mathbb{X}_j$ extracted from cropped square-level images. The learner selects a subset $A_j \subseteq [n_j]$ for full exposures. For a microscope, the exposure time $T_{\exp}$ and the latency $T_{\mathrm{mov}}$ for moving to the next patch are typically known or easily estimated. Let $t_j$ denote the stochastic time to acquire the square-level view and extract contexts $\mathbb{X}_j$. Then, the latency of action $A_j$ is

$$l_j(A_j) = t_j + T_{\mathrm{mov}}\mathbf{1}\{A_j \neq \emptyset\} + T_{\exp}|A_j|. \tag{1}$$

The feedback $y_{j,i}$ is obtained by evaluating the high-magnification micrographs. Micrograph quality can be quantified using the CTF maximum resolution (Rohou & Grigorieff, 2015), which measures the finest structural detail in Å (0.1 nm). With sufficient computational resources, additional metrics—such as the number of biomolecules detected per micrograph or assessments from deep learning models like MicAssess (Li et al., 2020)—can also be incorporated.

## 4 MAXIMIZATION OF AVERAGE REWARD

In this section, we study the maximum average reward achievable in the latency-aware contextual bandit problem. Assuming a known mean reward function $\psi_*$, we derive the Bellman optimality equation, which can be used to compute this maximum average reward. This quantity then serves as a baseline for defining the regret of an algorithm, which we aim to minimize throughout the paper.

### 4.1 OPTIMAL AVERAGE REWARD

At each round $k$, the learner follows a policy $\pi$ to select a subset of arms $A_k \in \mathbb{A}_k$. Let the *history* up to round $k$ be $\mathcal{H}_{k-1} \triangleq \left\{\left(\mathbb{X}_j, \mathbb{A}_j, l_j, \{y_{j,i}\}_{i \in A_j}\right)\right\}_{j=1}^{k-1}$. The policy $\pi$ maps the history $\mathcal{H}_{k-1}$ and the current arm features $\mathbb{X}_k$ to a probability distribution over the action space $\mathbb{A}_k$. Let $k(t)$ denote the (random) number of completed decision rounds up to time $t$. The *expected cumulative reward* of a policy $\pi$ up to time $t$ is

$$\mathbb{E}\left[\sum_{j=1}^{k(t)} \sum_{i \in A_j} y_{j,i}\right] = \mathbb{E}\left[\sum_{j=1}^{k(t)} \sum_{i \in A_j} \mathbb{E}[y_{j,i} \mid \mathcal{H}_{j-1}]\right] = \mathbb{E}\left[\sum_{j=1}^{k(t)} \sum_{i \in A_j} \psi_*(\mathbf{x}_{j,i})\right] \triangleq \mathbb{E}[q^\pi(t)], \tag{2}$$

which depends on both the environment $\mathcal{M}$ and the policy $\pi$. In the average-reward setting, the performance of $\pi$ is evaluated by the long-term average reward

$$\Gamma_{\mathcal{M}}^{\pi} \triangleq \limsup_{t \to \infty} \frac{\mathbb{E}[\mathsf{q}^{\pi}(t)]}{t}.$$

With Assumption 1 and the mean reward function $\psi_*$ bounded in $[-1, 1]$, $\Gamma_{\mathcal{M}}^{\pi}$ is finite for any policy $\pi$. Then *optimal average reward* is then defined as

$$\Gamma_{\mathcal{M}}^* \triangleq \sup_{\pi} \Gamma_{\mathcal{M}}^{\pi}.$$

Treating $\mathsf{l}_j(A_j)$ in round $j$ as the sojourn time, our model can be viewed as a special case of SMDPs (Puterman, 2014). The following theorem provides its Bellman optimality equation.

**Theorem 1.** *For the latency-aware contextual bandit problem $\mathcal{M} = (P_{\text{env}}, \psi_*)$, let $(\mathbb{X}, \mathbb{A}, \mathsf{l}) \sim P_{\text{env}}$ and let $\boldsymbol{\mu} = \{\mu_i\}_{i=1}^{\mathsf{n}}$, where $\mu_i = \psi_*(\mathbf{x}_i)$ for each $\mathbf{x}_i \in \mathbb{X}$. The optimal average reward $\Gamma = \Gamma_{\mathcal{M}}^*$ is the unique solution to $\mathbb{E}\big[\min_{A \in \mathbb{A}} g(\Gamma, A, \mathsf{l}, \boldsymbol{\mu})\big] = 0$, where*

$$g(\Gamma, A, \mathsf{l}, \boldsymbol{\mu}) \triangleq \mathsf{l}(A)\,\Gamma - \sum_{i \in A} \mu_i. \tag{3}$$

*Proof.* We adopt the concept of differential return from average-reward MDPs (Sutton, 2018). At eacg decision round $j$, selecting $A_j \in \mathbb{A}_j$ incurs time $\mathsf{l}_j(A_j)$. The quantity $g(\Gamma_{\mathcal{M}}^*, A_j, \mathsf{l}_j, \boldsymbol{\mu}_j)$ defined in equation 3 measures the *gap* between the optimal expected reward $\mathsf{l}_j(A_j)\Gamma_{\mathcal{M}}^*$ in the time interval and the actual collected reward $\sum_{i \in A_{j,i}} \mu_{j,i}$.

**Step 1:** For any policy $\pi$, let $A_j \in \mathbb{A}_j$ denote the selected arms at round $j$. Since $t$ is possible in the middle of a decision round, $\left| t - \sum_{j=1}^{\mathsf{k}(t)} \mathsf{l}_j(A_j) \right| \leq l_{\max}$, where $l_{\max}$ is from Assumption 1. Then

$$t\Gamma_{\mathcal{M}}^* - \mathsf{q}^{\pi}(t) \geq \sum_{j=1}^{\mathsf{k}(t)} g(\Gamma_{\mathcal{M}}^*, A_j, \mathsf{l}_j, \boldsymbol{\mu}_j) - l_{\max}|\Gamma_{\mathcal{M}}^*| \geq \sum_{j=1}^{\mathsf{k}(t)} \min_{A \in \mathbb{A}_j} g(\Gamma_{\mathcal{M}}^*, A, \mathsf{l}_j, \boldsymbol{\mu}_j) - l_{\max}|\Gamma_{\mathcal{M}}^*|. \tag{4}$$

Using Wald's lemma (Durrett, 2019) and the IID assumption on $\{(\mathbb{X}_j, \mathbb{A}_j, \mathsf{l}_j)\}_{j=1}^{\infty}$,

$$\mathbb{E}\left[ \sum_{j=1}^{\mathsf{k}(t)} \min_{A \in \mathbb{A}_j} g(\Gamma_{\mathcal{M}}^*, A, \mathsf{l}_j, \boldsymbol{\mu}_j) \right] = \mathbb{E}[\mathsf{k}(t)]\, \mathbb{E}\left[ \min_{A \in \mathbb{A}} g(\Gamma_{\mathcal{M}}^*, A, \mathsf{l}, \boldsymbol{\mu}) \right]. \tag{5}$$

With equation 5, dividing equation 4 by $t$ and taking the $\limsup$ of expectations yields

$$\Gamma_{\mathcal{M}}^* - \Gamma_{\mathcal{M}}^{\pi} \geq \limsup_{t \to \infty} \frac{\mathbb{E}[\mathsf{k}(t)]}{t} \mathbb{E}\left[ \min_{A \in \mathbb{A}} g(\Gamma_{\mathcal{M}}^*, A, \mathsf{l}, \boldsymbol{\mu}) \right].$$

Since this holds for any $\pi$, taking the supremum of $\Gamma_{\mathcal{M}}^{\pi}$ over $\pi$ gives

$$\limsup_{t \to \infty} \frac{\mathbb{E}[\mathsf{k}(t)]}{t} \mathbb{E}\left[ \min_{A \in \mathbb{A}} g(\Gamma_{\mathcal{M}}^*, A, \mathsf{l}, \boldsymbol{\mu}) \right] \leq 0.$$

Since $\limsup_{t \to \infty} \mathbb{E}[\mathsf{k}(t)]/t > 0$, we get $\mathbb{E}\big[\min_{A \in \mathbb{A}} g(\Gamma_{\mathcal{M}}^*, A, \mathsf{l}, \boldsymbol{\mu})\big] \leq 0$.

**Step 2:** Let policy $\pi'$ selects $A_j = \arg\min_{A \in \mathbb{A}_j} g(\Gamma_{\mathcal{M}}^*, A, \mathsf{l}_j, \boldsymbol{\mu}_j)$ at each round $j$. Using the same bound for $t$ in the middle of a decision round,

$$t\Gamma_{\mathcal{M}}^* - \mathsf{q}^{\pi'}(t) \leq \sum_{j=1}^{\mathsf{k}(t)} \min_{A \in \mathbb{A}_j} g(\Gamma_{\mathcal{M}}^*, A, \mathsf{l}_j, \boldsymbol{\mu}_j) + l_{\max}|\Gamma_{\mathcal{M}}^*|\,.$$

Taking expectations and applying equation 5 give

$$\limsup_{t \to \infty} \frac{\mathbb{E}[\mathsf{k}(t)]}{t} \mathbb{E}\left[ \min_{A \in \mathbb{A}} g(\Gamma_{\mathcal{M}}^*, A, \mathsf{l}, \boldsymbol{\mu}) \right] \geq \Gamma_{\mathcal{M}}^* - \Gamma_{\mathcal{M}}^{\pi'} \geq 0.$$

---

**Algorithm 1:** Contextual Online Arm Filtering (COAF)

---

**Initialization:** $\xi \in (0, 1]$, $\Gamma_1 \in [\Gamma_{\min}, \Gamma_{\max}]$ and $\gamma_0 = 0$.

**for** $j = 1, 2, \ldots$ **do**

1     Estimate the mean rewards for each arm $i$ base on $\mathcal{H}_{j-1}$, denoted by $\hat{\boldsymbol{\mu}}_j = \{\hat{\mu}_{j,i}\}_{i=1}^{n_j}$.

2     Select subset of arms
$$A_j \in \underset{A \in \mathbb{A}_j}{\arg\min}\, g(\Gamma_j, A, 1_j, \hat{\boldsymbol{\mu}}_j).$$

3     Set $\gamma_j = \gamma_{j-1} + \min_{A \in \mathbb{A}_j} 1_j(A)$, and set $\Gamma_{j+1} = \Pi_{[\Gamma_{\min}, \Gamma_{\max}]} \left[ \Gamma_j - \frac{1}{\xi \gamma_j} g(\Gamma_j, A_j, 1_j, \hat{\boldsymbol{\mu}}_j) \right]$.

---

Since $\limsup_{t \to \infty} \mathbb{E}[k(t)]/t > 0$, we have $\mathbb{E}\left[ \min_{A \in \mathbb{A}} g(\Gamma_{\mathcal{M}}^*, A, 1, \boldsymbol{\mu}) \right] \geq 0$.

Combining both steps, we conclude $\mathbb{E}\left[ \min_{A \in \mathbb{A}} g(\Gamma_{\mathcal{M}}^*, A, 1, \boldsymbol{\mu}) \right] = 0$. With $1(A) > L_{\min} > 0$, the function $g(\Gamma, A, 1, \boldsymbol{\mu})$ is strictly increasing in $\Gamma$, and hence $\mathbb{E}\left[ \min_{A \in \mathbb{A}} g(\Gamma, A, 1, \boldsymbol{\mu}) \right]$ is also strictly increasing in $\Gamma$. Therefore, the solution $\Gamma = \Gamma_{\mathcal{M}}^*$ is unique. $\qquad\square$

**Remark 2.** *The effect of arm quality and latency on the optimal average reward $\Gamma_{\mathcal{M}}^*$ can be seen from Theorem 1. Larger delays $1(A)$ steepen the growth of $\mathbb{E}[\min_{A \in \mathbb{A}} g(\Gamma, A, 1, \boldsymbol{\mu})]$ with $\Gamma$, resulting in a smaller root $\Gamma_{\mathcal{M}}^*$. Conversely, higher mean rewards $\boldsymbol{\mu}$ shift the function downward, yielding a larger $\Gamma_{\mathcal{M}}^*$. The proof also indicates that the policy $\pi'$ minimizing $g(\Gamma_{\mathcal{M}}^*, A, 1_j, \boldsymbol{\mu}_j)$ is optimal.*

### 4.2 ALGORITHM REGRET

Since the objective of a policy $\pi$ is to maximize cumulative reward, the optimal average reward $\Gamma_{\mathcal{M}}^*$ serves as a natural performance benchmark. The *regret* of $\pi$ at time $T$ is defined as

$$R_T^\pi \triangleq T\Gamma_{\mathcal{M}}^* - \mathbb{E}\left[ q^\pi(T) \right], \tag{6}$$

which we aim to minimize. Our goal is to design policies that perform well across general problem setups. Specifically, policy $\pi$ aims to minimize the *worst-case regret* $\sup_{\mathcal{M}} R_T^\pi(\mathcal{M})$, while the optimal value of this quantity, known as the *minimax regret*, is given by $\inf_\pi \sup_{\mathcal{M}} R_T^\pi(\mathcal{M})$.

The minimax regret lower bound in contextual bandit settings is known to depend on the regressor class $\mathcal{F}$. Since contextual bandits are a special case of the problem studied here, these lower bound results also apply. In particular, when $\mathcal{F}$ is the class of $d$-dimensional linear functions, the state-of-the-art minimax regret lower bound is $\Omega(d\sqrt{T})$ (Lattimore & Szepesvári, 2020).

## 5 CONTEXTUAL ONLINE ARM FILTERING ALGORITHM

The latency-aware contextual bandit problem poses a significant challenge, as it requires learning both the mean reward function $\psi_*$ and the potentially complex distribution $P_{\text{env}}$ underlying for $\{(\mathbb{X}_j, \mathbb{A}_j, 1_j)\}_{j=1}^\infty$. A key insight from the proof of Theorem 1 (step 2) is that it is optimal to take action $A_j \in \arg\min_{A \in \mathbb{A}_j} g(\Gamma_{\mathcal{M}}^*, A, 1_j, \boldsymbol{\mu}_j)$ in each round $j$. Since $\Gamma_{\mathcal{M}}^*$ and $\boldsymbol{\mu}_j$ are unknown, this minimization is not directly feasible. In this section, we introduce and analyze the COAF algorithm, which relies on estimated values of $\Gamma_{\mathcal{M}}^*$ and $\boldsymbol{\mu}_j$ to filter out suboptimal arms.

### 5.1 A GENERIC ALGORITHM FRAMEWORK

Stochastic approximation (Robbins & Monro, 1951) is a standard method for finding the root of an unknown real-valued function using noisy measurements. Since in Theorem 1 has shown that $\mathbb{E}\left[ \min_{A \in \mathbb{A}} g(\Gamma_{\mathcal{M}}^*, A, 1, \boldsymbol{\mu}) \right] = 0$, COAF in Algorithm 1 employs this approach to maintain an estimator $\Gamma_j$ for $\Gamma_{\mathcal{M}}^*$ at each round $j$. In Line 3 of Algorithm 1, $\Gamma_j$ is updated via stochastic approximation and projected back to $[\Gamma_{\min}, \Gamma_{\max}]$, where $\Gamma_{\min} = -n_{\max}/l_{\min}$ and $\Gamma_{\max} = n_{\max}/l_{\min}$.

Since the true reward function $\psi_*$ is unknown, COAF estimates the mean rewards $\hat{\boldsymbol{\mu}}_j$ from the sampling history $\mathcal{H}_{j-1}$. To balance exploration and exploitation, we adopt a UCB strategy by constructing a confidence set $\mathcal{F}_j \subseteq \mathcal{F}$ based on $\mathcal{H}_{j-1}$, which contains $\psi_*$ with high probability. At

each round $j$, the UCB for arm $i$ is $\hat{\mu}_{j,i} = \max_{\psi \in \mathcal{F}_j} \psi(\mathbf{x}_{j,i})$, ensuring, with high probability, that $\hat{\mu}_{j,i} \geq \mu_{j,i} = \psi_*(\mathbf{x}_{j,i})$. In this paper, we focus on the UCB approach due to its simplicity and theoretical soundness. However, it is plausible to expect that other index-based contextual bandit algorithms, such as Thompson Sampling, can also be adapted to the COAF framework.

## 5.2 WORST-CASE REGRET ANALYSIS FOR COAF

In Algorithm 1, if $\hat{\boldsymbol{\mu}}_j = \boldsymbol{\mu}_j$, the stochastic approximation procedure ensures $\mathbb{E}\big[(\Gamma_j - \Gamma_{\mathcal{M}}^*)^2\big] \to 0$ as $j \to \infty$. The following result is derived by relating the regret to the convergence rate of $\Gamma_j$.

**Lemma 2.** *Consider any latency-aware contextual bandit problem $\mathcal{M} = (P_{\text{env}}, \psi_*)$. Suppose the COAF algorithm runs with $\xi = 1$ and has exact mean reward estimates, i.e., $\hat{\boldsymbol{\mu}}_j = \boldsymbol{\mu}_j$ in every round $j$. Then, for any time horizon $T > 0$, the regret of COAF satisfies*

$$R_T^{\text{C}} \leq \sqrt{U_T} + \frac{n_{\max} l_{\max}}{l_{\min}}, \text{ where } U_T \triangleq \frac{T}{l_{\min}} \left(\frac{l_{\max} n_{\max}}{l_{\min}}\right)^2 \left(1 + \frac{l_{\max}}{l_{\min}}\right)^2 \left[1 + \log\left(\frac{T}{l_{\min}}\right)\right].$$

**Remark 3.** *Lemma 2 captures the oracle case where COAF has access to the true mean rewards, and establishes an $O(\sqrt{T \log T})$ regret upper bound that arises solely from learning $\Gamma_{\mathcal{M}}^*$.*

The general COAF algorithm needs to learn the unknown reward function $\psi_*$ within the regressor class $\mathcal{F}$ from noisy observations. Following standard practice in the contextual bandit literature (Lattimore & Szepesvári, 2020), we assume that the noise is conditionally sub-Gaussian.

**Assumption 3.** *For any $j \in \mathbb{N}$, $\{\epsilon_{j,i}\}_{i=1}^{n_j}$ are independent and conditionally 1-subgaussian:*

$$\mathbb{E}\left[e^{\alpha \epsilon_{j,i}} \mid \mathcal{H}_{j-1}\right] \leq \exp\left(\frac{\alpha^2}{2}\right), \quad \forall \alpha \in \mathbb{R}, \ \forall i \in [n_j].$$

### 5.2.1 REGRET UPPER BOUND WITH LINEAR $\mathcal{F}$

The linear regressor class is defined as $\mathcal{F}^l \triangleq \big\{\boldsymbol{x} \mapsto \langle \theta, \boldsymbol{x} \rangle \mid \theta \in \mathbb{R}^d, \ \|\theta\| \leq 1\big\}$, where the context space is $\Omega^l \triangleq \big\{\boldsymbol{x} \in \mathbb{R}^d \mid \|\boldsymbol{x}\| \leq 1\big\}$. To estimate $\theta_*$ corresponding to $\psi_*$, let

$$\bar{\theta}_k = \boldsymbol{V}_k^{-1}(\lambda) \sum_{j=1}^{k} \sum_{i \in A_j} \mathbf{x}_{j,i} \mathbf{y}_{j,i}, \ \boldsymbol{V}_k(\lambda) = \lambda \boldsymbol{I} + \sum_{j=1}^{k} \sum_{i \in A_j} \mathbf{x}_{j,i} \mathbf{x}_{j,i}^{\top},$$

where $\lambda > 0$ is the regularization parameter. The regressor confidence set at round $k$ is defined as

$$\mathcal{F}_k \triangleq \left\{\boldsymbol{x} \mapsto \langle \theta, \boldsymbol{x} \rangle \ \Big| \ \theta \in \mathbb{R}^d, \ \big\|\theta - \bar{\theta}_{k-1}\big\|_{\boldsymbol{V}_{k-1}(\lambda)}^2 \leq \beta(\mathbf{s}_{k-1}, \delta), \ \|\theta\| \leq 1\right\},$$

where $\delta \in (0, 1)$, $\mathbf{s}_k = \sum_{j=1}^{k} |A_j|$ and $\sqrt{\beta(n, \delta)} = \sqrt{\lambda} + \sqrt{2 \log(\frac{1}{\delta}) + d \log\left(\frac{d\lambda + n}{d\lambda}\right)}$.

**Theorem 3.** *For any latency-aware contextual bandit problem $\mathcal{M} = (P_{\text{env}}, \psi_*)$, suppose $\psi_* \in \mathcal{F}^l$ and the context space is $\Omega^l$. For any $\delta \in (0, 1/\sqrt{e}]$, with probability at least $1 - \delta$, the regret of COAF with parameter $\xi \in (0, 1)$ at any time $T > 0$ satisfies*

$$R_T^{\text{C}} \leq \sqrt{\frac{U_T}{\xi} + \frac{1}{1 - \xi} \left(\frac{l_{\max}}{l_{\min}}\right)^3 W_T(\delta)} + \sqrt{\frac{n_{\max}}{l_{\min}} W_T(\delta)} + \frac{n_{\max} l_{\max}}{l_{\min}},$$

*where $W_T(\delta) = 8dT\beta\left(\frac{Tn_{\max}}{l_{\min}}, \delta\right) \log\left(\frac{d\lambda l_{\min} + Tn_{\max}}{d\lambda l_{\min}}\right)$.*

### 5.2.2 REGRET UPPER BOUND WITH GENERAL $\mathcal{F}$

For a general regressor class $\mathcal{F}$, let $N(\mathcal{F}, \alpha, \|\cdot\|_\infty)$ be the $\alpha$-covering number of $\mathcal{F}$ in the sup-norm $\|\cdot\|_\infty$. By calling the online regression oracle, COAF computes the estimated regressor

$$\bar{\psi}_k \in \arg\min_{\psi \in \mathcal{F}} \sum_{j=1}^{k} \sum_{i \in A_j} \left[\psi(\mathbf{x}_{j,i}) - \mathbf{y}_{j,i}\right]^2.$$

The abstract confidence set $\mathcal{F}_k$ centers around $\bar{\psi}_k$ and is defined as

$$\mathcal{F}_k \triangleq \left\{ \psi \in \mathcal{F} \;\Big|\; \sum_{j=1}^{k-1} \sum_{i \in A_j} [\psi(\mathbf{x}_{j,i}) - \bar{\psi}_{k-1}(\mathbf{x}_{j,i})]^2 \leq \tilde{\beta}(s_{k-1}, \mathcal{F}, \delta, \alpha) \right\},$$

where $\tilde{\beta}(n, \mathcal{F}, \delta, \alpha) \triangleq 8 \log \left( N(\mathcal{F}, \alpha, \|\cdot\|_\infty)/\delta \right) + 2\alpha n \big( 8 + \sqrt{8 \log(4n^2/\delta)} \big)$ is the tolerence. In practice, $\alpha$ can be chosen on the order of $1/T$.

The eluder dimension of a function class $\mathcal{F}$, defined below, measures reward dependencies across contexts (Russo & Van Roy, 2013) and is widely used in contextual bandit regret analysis.

**Definition 1.** *Feature $\boldsymbol{x}$ is $\epsilon$-dependent on $\{\boldsymbol{x}_1, \ldots, \boldsymbol{x}_n\}$ with respect to $\mathcal{F}$ if any pair of functions $\psi, \psi' \in \mathcal{F}$ satisfying $\sqrt{\sum_{k=1}^{n} [\psi(\boldsymbol{x}_k) - \psi'(\boldsymbol{x}_k)]^2} \leq \epsilon$ also satisfies $\psi(\boldsymbol{x}) - \psi'(\boldsymbol{x}) \leq \epsilon$. Furthermore, $\boldsymbol{x}$ is $\epsilon$-independent of $\{\boldsymbol{x}_1, \ldots, \boldsymbol{x}_n\}$ with respect to $\mathcal{F}$ if $\boldsymbol{x}$ is not $\epsilon$-dependent on $\{\boldsymbol{x}_1, \ldots, \boldsymbol{x}_n\}$.*

**Definition 2.** *The $\epsilon$-eluder dimension $\dim_{\mathrm{E}}(\mathcal{F}, \epsilon)$ is the length of the longest sequence of elements in $\Omega$ such that, for some $\epsilon' > \epsilon$, every element is $\epsilon'$-independent of its predecessors.*

**Theorem 4.** *Consider latency-aware contextual bandit problem $\mathcal{M} = (P_{\mathrm{env}}, \psi_*)$. Suppose $\psi_* \in \mathcal{F}$ and let $d_T = \dim_{\mathrm{E}}\left(\mathcal{F}, \frac{l_{\min}}{T n_{\max}}\right)$. For any $\delta > 0$, with probability at least $1 - 2\delta$, the regret of COAF with parameter $\xi \in (0, 1)$ at any time $T$ satisfies*

$$R_T^{\mathrm{C}} \leq \sqrt{\frac{U_T}{\xi} + \frac{1}{1-\xi}\left(\frac{l_{\max}}{l_{\min}}\right)^3 \tilde{W}_T(\delta)} + \sqrt{\frac{n_{\max}}{l_{\min}} \tilde{W}_T(\delta)} + \frac{n_{\max} l_{\max}}{l_{\min}},$$

*where $\tilde{W}_T(\delta)/T = 4d_T + 1 + 4\left[\tilde{\beta}\left(\frac{T}{n_{\min}}, \mathcal{F}, \delta, \alpha\right) + 4n_{\max}\right] d_T \left(1 + \log\left(\frac{T n_{\max}}{l_{\min}}\right)\right).$*

**Remark 4.** *In Theorems 3 and 4, the regret upper bounds introduce new terms $W_T(\delta)$ and $\tilde{W}_T(\delta)$, which arise from learning a regressor in $\mathcal{F}$. The parameter $\xi$ controls the trade-off between learning $\Gamma_{\mathcal{M}}^*$ and learning the regressor. By setting $\delta = \alpha = 1/T$, the regret upper bounds for COAF become $O(d\sqrt{T}\log T)$ and $O\left(\sqrt{\dim_E\left(\mathcal{F}, \frac{1}{T}\right) \log\left(N(\mathcal{F}, \frac{1}{T}, \|\cdot\|_\infty)\right) T \log T}\right)$. These rates are consistent with the standard contextual bandit literature (Chu et al., 2011; Russo & Van Roy, 2013).*

## 6 NUMERICAL EXPERIMENTS

In this section, we evaluate the performance of COAF in two experiments. In the movie recommendation task, we compare its regret against three baseline algorithms. In the cryo-EM data collection task, we assess its data collection efficiency against human microscopists.

### 6.1 SIMULATIONS WITH MOVIELENS 1M DATA

We use the MovieLens 1M dataset (Harper & Konstan, 2015), which contains ratings for 3461 movies by 6040 users. Missing ratings are predicted via a matrix completion technique (Nie et al., 2012), and principal component analysis is applied to obtain a 10-dimensional feature vector for each movie. At each round $j$, the number of available movies $n_j$ is uniformly sampled from 6 to 20, and the action space $\mathbb{A}_j = 2^{[n_j]}$. The time cost is $l_j(A_j) = t_j + |A_j|$, where $t_j$ is uniformly sampled from 5 to 10. The ground-truth regressor $\psi_*$ is trained on the average ratings across all users, and the noisy reward for each movie is given by the rating from a randomly selected user.

In our experiment, the optimal average reward $\Gamma_{\mathcal{M}}^*$ is computed via prolonged execution of the stochastic approximation algorithm (Robbins & Monro, 1951). Each algorithm in Fig. 2 is run for 2000 iterations, and we plot the mean regret along with the 5% lower, 90% middle, and 5% upper quantiles of empirical regrets in Fig. 2. In COAF-TS, we follow Abeille & Lazaric (2017) and sample $\hat{\theta}_j$ from $\mathcal{N}(\bar{\theta}_{j-1}, \sqrt{\beta(s_{j-1}, \delta)} V_{j-1}(\lambda))$ as the parameter for $\hat{\psi}_j$ to estimate arm rewards in round $j$. In Fig. 2d, a fixed threshold of 1.75 fails to achieve sublinear regret. In Fig. 2a, COAF-ORC corresponds to the oracle case described in Lemma 2, where the regret reflects only the cost of learning $\Gamma_{\mathcal{M}}^*$. For both COAF and COAF-TS, we set the parameter $\xi = 0.5$. We further observe that the original COAF with UCB estimates outperforms its Thompson sampling variant.

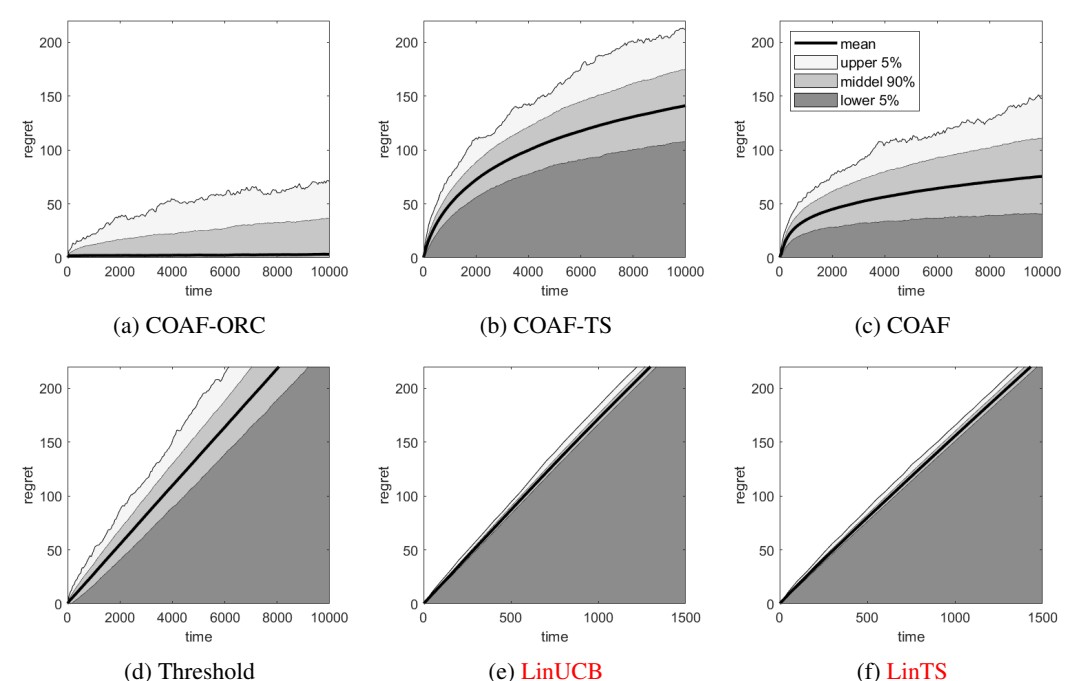

Figure 2: Regrets of COAF and 3 baselines: (i) Threshold selects movies with rating $\geq 1.75$, (ii) COAF-ORC uses the true mean reward, and (iii) COAF-TS employs Thompson Sampling approach.

## 6.2 CRYO-EM DATA COLLECTION SIMULATIONS

We evaluate COAF in a realistic cryo-EM setting by benchmarking its automated data collection performance against human microscopists. To simulate experimental conditions, we use two existing datasets. The microscope parameters $t_j$, $T_{\mathrm{mov}}$, and $T_{\mathrm{exp}}$ in equation 1 are sampled from real experiment logs, with mean values of 12.09, 51.99, and 6.66 seconds, respectively.

The data collection simulator is illustrated in Fig. 3a. In this setup, holes are cropped from medium-magnification images. The feature representation of each hole consists of its mean pixel value, which correlates with ice thickness, together with the output of the deep learning model Ptolemy (Kim et al., 2023). Based on these features, the UCB reward estimate is displayed at the top-right corner of each hole. In the first experiment ( Fig. 3b), we use 0/1 reward feedback, where a micrograph is labeled good if its CTF maximum resolution is below 3.8 Å. In the second experiment ( Fig. 3c), the reward corresponds to the number of particles in the micrograph, estimated via blob picking. We then reconstruct 3D biomolecular structures with the same computation method using data collected by human microscopists and by COAF. In both cases, COAF improves data collection efficiency over human microscopists, yielding more good micrographs (particles) and higher-resolution 3D structures. These results highlight the promise of COAF for cryo-EM automation.

## 7 CONCLUSION

In this paper, we studied the latency-aware contextual bandit problem, which generalizes both standard contextual bandits and mortal multi-armed bandits to settings where collecting contextual information and taking actions incur time costs. We formulated the problem as a special case of SMDPs with unknown rewards, sojourn times, and transition dynamics, and defined regret relative to an optimal algorithm that maximizes the long-term average reward. Building on stochastic approximation and UCB methods, we proposed the COAF algorithm and established its regret guarantees. Through simulations on movie recommendation tasks and cryo-EM data collection, we demonstrated that COAF efficiently maximizes cumulative reward over time. Our approach provides a

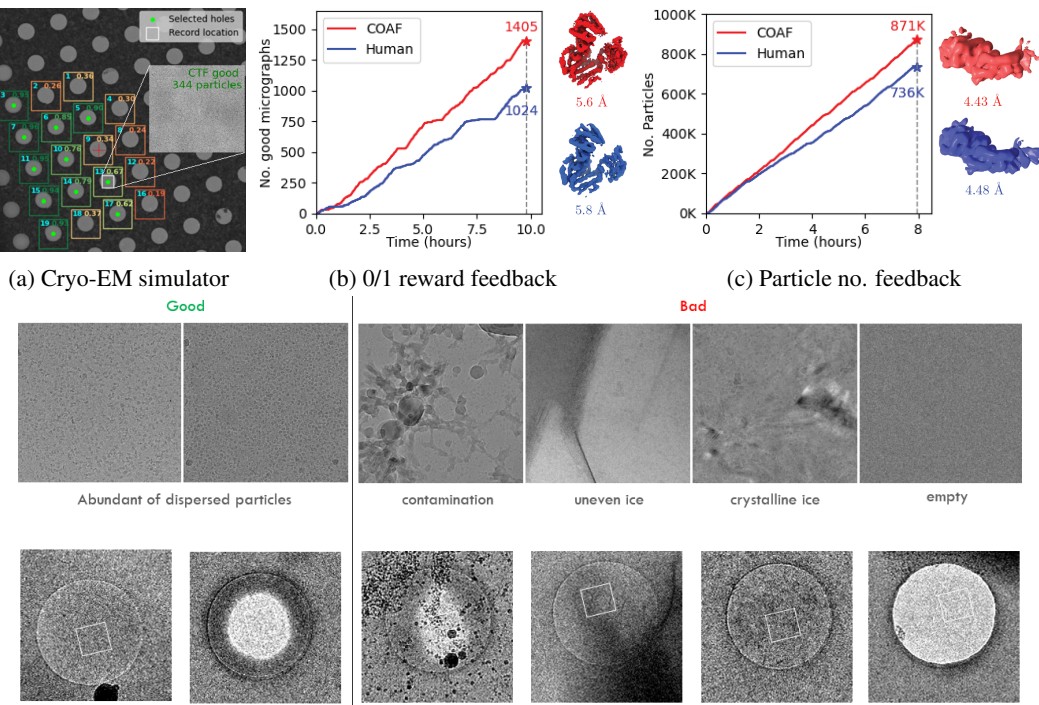

(a) Cryo-EM simulator   (b) 0/1 reward feedback   (c) Particle no. feedback

(d) Correlation between medium-magnification holes and micrograph qualities

Figure 3: Setup and results of the cryo-EM data collection experiment.

fully automated pipeline for cryo-EM data collection and shows promise for broader application in other time-sensitive decision-making scenarios.

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

## A    REGRET OF COAF WITH ACCESS TO TRUE MEAN REWARDS

*Proof of Lemma 2.* With the regret defined in equation 6, COAF satisfies

$$R_T^{\mathrm{C}} = T\Gamma_{\mathcal{M}}^* - \mathbb{E}\left[\sum_{j=1}^{\mathrm{k}(T)} \sum_{i \in A_j} \mu_{j,i}\right]$$

$$\leq \mathbb{E}\left[\sum_{j=1}^{\mathrm{k}(T)} \left(\mathrm{l}_j(A_j)\Gamma_{\mathcal{M}}^* - \sum_{i \in A_j} \mu_{j,i}\right)\right] + \underbrace{l_{\max}\Gamma_{\mathcal{M}}^*}_{(a)}$$

$$\leq \mathbb{E}\left[\sum_{j=1}^{\mathrm{k}(T)} g(\Gamma_{\mathcal{M}}^*, A_j, \mathrm{l}_j, \boldsymbol{\mu}_j)\right] + l_{\max}\Gamma_{\max}, \tag{7}$$

where term $(a)$ accounts for the potentially unfinished decision round. Note that at each round $j$, with access to the true mean rewards $\boldsymbol{\mu}_j$, COAF selects a subset of arms

$$A_j \in \arg\min_{A \in \mathbb{A}_j} g(\Gamma_j, A, \mathrm{l}_j, \boldsymbol{\mu}_j), \tag{8}$$

where $\Gamma_j$ is the online estimator of $\Gamma_{\mathcal{M}}^*$.

**Step 1:** In this step, we derive an upper bound for $g(\Gamma_{\mathcal{M}}^*, A_j, \mathrm{l}_j, \boldsymbol{\mu}_j)$ at each round $j$. Let

$$A_j^* \in \arg\min_{A \in \mathbb{A}_j} g(\Gamma_{\mathcal{M}}^*, A, \mathrm{l}_j, \boldsymbol{\mu}_j).$$

We consider two cases separately: $\Gamma_j < \Gamma_{\mathcal{M}}^*$ and $\Gamma_j \geq \Gamma_{\mathcal{M}}^*$.

*Case 1:* If $\Gamma_j < \Gamma_{\mathcal{M}}^*$, we have

$$g(\Gamma_{\mathcal{M}}^*, A_j, \mathrm{l}_j, \boldsymbol{\mu}_j) = \mathrm{l}_j(A_j)(\Gamma_{\mathcal{M}}^* - \Gamma_j) + \mathrm{l}_j(A_j)\Gamma_j - \sum_{i \in A_j} \mu_{j,i}$$

$$\text{according to equation 8}$$

$$\leq \mathrm{l}_j(A_j)(\Gamma_{\mathcal{M}}^* - \Gamma_j) + \mathrm{l}_j(A_j^*)\Gamma_j - \sum_{i \in A_j^*} \mu_{j,i}$$

$$\leq \mathrm{l}_j(A_j)(\Gamma_{\mathcal{M}}^* - \Gamma_j) + \mathrm{l}_j(A_j^*)\Gamma_{\mathcal{M}}^* - \sum_{i \in A_j^*} \mu_{j,i}$$

$$= \mathrm{l}_j(A_j)(\Gamma_{\mathcal{M}}^* - \Gamma_j) + g(\Gamma_{\mathcal{M}}^*, A_j^*, \mathrm{l}_j, \boldsymbol{\mu}_j).$$

*Case 2:* If $\Gamma_j \geq \Gamma_{\mathcal{M}}^*$, we have

$$g(\Gamma_{\mathcal{M}}^*, A_j, \mathrm{l}_j, \boldsymbol{\mu}_j) \leq \mathrm{l}_j(A_j)\Gamma_j - \sum_{i \in A_j} \mu_{j,i}$$

$$\text{according to equation 8}$$

$$\leq \mathrm{l}_j(A_j^*)\Gamma_j - \sum_{i \in A_j^*} \mu_{j,i}$$

$$= \mathrm{l}_j(A_j^*)(\Gamma_j - \Gamma_{\mathcal{M}}^*) + \mathrm{l}_j(A_j^*)\Gamma_{\mathcal{M}}^* - \sum_{i \in A_j^*} \mu_{j,i}$$

$$= \mathrm{l}_j(A_j^*)(\Gamma_j - \Gamma_{\mathcal{M}}^*) + g(\Gamma_{\mathcal{M}}^*, A_j^*, \mathrm{l}_j, \boldsymbol{\mu}_j).$$

Putting both cases together, we have

$$g(\Gamma_{\mathcal{M}}^*, A_j, \mathrm{l}_j, \boldsymbol{\mu}_j) \leq l_{\max}|\Gamma_j - \Gamma_{\mathcal{M}}^*| + g(\Gamma_{\mathcal{M}}^*, A_j^*, \mathrm{l}_j, \boldsymbol{\mu}_j).$$

From Theorem 1, we also have $\mathbb{E}[g(\Gamma_{\mathcal{M}}^*, A_j^*, \mathrm{l}_j, \boldsymbol{\mu}_j)] = 0$. Applying this result to equation 7 yields

$$R_T^{\mathrm{C}} \leq \mathbb{E}\left[\sum_{j=1}^{\mathrm{k}(T)} l_{\max}|\Gamma_j - \Gamma_{\mathcal{M}}^*|\right] + l_{\max}\Gamma_{\max}. \tag{9}$$

**Step 2:** With equation 9, we have converted bounding the regret to analyzing the convergence of $\Gamma_j$. To ease the notation, we define

$$h_j(\Gamma) \triangleq \min_{A \in \mathbb{A}_j} g(\Gamma, A, 1_j, \boldsymbol{\mu}_j) = \min_{A \in \mathbb{A}_j} \left[ 1_j(A)\,\Gamma - \sum_{i \in A} \mu_i \right]. \tag{10}$$

Let $f_j(\Gamma)$ be a function such that $f_j'(\Gamma) = h_j(\Gamma)$. Notice that $f_j''(\Gamma) = h_j'(\Gamma) \geq \min_{A \in \mathbb{A}_j} 1_j(A)$, which means $f_j$ is strongly convex with parameter $c_j = \min_{A \in \mathbb{A}_j} 1_j(A)$. So we have

$$f_j(\Gamma_j) \geq f_j(\Gamma_{\mathcal{M}}^*) + (\Gamma_j - \Gamma_{\mathcal{M}}^*)h_j(\Gamma_{\mathcal{M}}^*) + \frac{c_j}{2}(\Gamma_j - \Gamma_{\mathcal{M}}^*)^2. \tag{11}$$

Applying Cauchy–Schwarz inequality, we have

$$\left( \sum_{j=1}^{k(T)} l_{\max} |\Gamma_j - \Gamma_{\mathcal{M}}^*| \right)^2 = \left( \sum_{j=1}^{k(T)} \frac{l_{\max}}{\sqrt{c_j}} \sqrt{c_j} |\Gamma_j - \Gamma_{\mathcal{M}}^*| \right)^2$$

$$\leq \left[ \sum_{j=1}^{k(T)} \left( \frac{l_{\max}}{\sqrt{c_j}} \right)^2 \right] \left[ \sum_{j=1}^{k(T)} c_j (\Gamma_j - \Gamma_{\mathcal{M}}^*)^2 \right]$$

$$\text{since } c_j \geq l_{\min}$$

$$\leq \left[ \sum_{j=1}^{k(T)} \frac{l_{\max}^2}{l_{\min}} \right] \left[ \sum_{j=1}^{k(T)} c_j (\Gamma_j - \Gamma_{\mathcal{M}}^*)^2 \right]$$

$$\text{since } k(T) \leq T/l_{\min}$$

$$\leq T \left( \frac{l_{\max}}{l_{\min}} \right)^2 \left[ \sum_{j=1}^{k(T)} c_j (\Gamma_j - \Gamma_{\mathcal{M}}^*)^2 \right]. \tag{12}$$

We substitute equation 11 into equation 12 to get

$$\left( \sum_{j=1}^{k(T)} l_{\max} |\Gamma_j - \Gamma_{\mathcal{M}}^*| \right)^2 \leq 2T \left( \frac{l_{\max}}{l_{\min}} \right)^2 \sum_{j=1}^{k(T)} \left[ f_j(\Gamma_j) - f_j(\Gamma_{\mathcal{M}}^*) + (\Gamma_{\mathcal{M}}^* - \Gamma_j)h_j(\Gamma_{\mathcal{M}}^*) \right]. \tag{13}$$

**Step 3:** In this step, we derive an upper bound on $\sum_{j=1}^{k(T)} f_j(\Gamma_j) - f_j(\Gamma_{\mathcal{M}}^*)$ from equation 13. We follow a standard procedure in analyzing online gradient descent for strongly convex functions (Hazan et al., 2016, Sec. 3.3). We reproduce it here to make the proof self-contained. Since $f_j'(\Gamma) = h_j(\Gamma)$ and $f_j$ is strongly convex with parameter $c_j$, we have

$$2 \left[ f_j(\Gamma_j) - f_j(\Gamma_{\mathcal{M}}^*) \right] \leq 2 h_j(\Gamma_j)(\Gamma_j - \Gamma_{\mathcal{M}}^*) - c_j(\Gamma_j - \Gamma_{\mathcal{M}}^*)^2. \tag{14}$$

With the update rule of $\Gamma_j$ in Algorithm 1, we apply the fact $g(\Gamma_j, A, 1_j, \boldsymbol{\mu}_j) = h_j(\Gamma_j)$ to get

$$(\Gamma_{j+1} - \Gamma_{\mathcal{M}}^*)^2 = \left[ \Pi_{[\Gamma_{\min}, \Gamma_{\max}]} \left( \Gamma_j - \frac{1}{\gamma_j} h_j(\Gamma_j) \right) - \Gamma_{\mathcal{M}}^* \right]^2 \leq \left[ \Gamma_j - \frac{1}{\gamma_j} h_j(\Gamma_j) - \Gamma_{\mathcal{M}}^* \right]^2$$

$$= (\Gamma_j - \Gamma_{\mathcal{M}}^*)^2 + \frac{1}{\gamma_j^2} h_j^2(\Gamma_j) - \frac{2}{\gamma_j} h_j(\Gamma_j)(\Gamma_j - \Gamma_{\mathcal{M}}^*).$$

Rearranging the above inequality, we get

$$2 h_j(\Gamma_j)(\Gamma_j - \Gamma_{\mathcal{M}}^*) \leq \gamma_j (\Gamma_j - \Gamma_{\mathcal{M}}^*)^2 - \gamma_j (\Gamma_{j+1} - \Gamma_{\mathcal{M}}^*)^2 + \frac{1}{\gamma_j} h_j^2(\Gamma_j). \tag{15}$$

Substituting equation 14 into equation 15 and summing from $j = 1$ to $k(T)$, we have

$$2\sum_{j=1}^{k(T)} f_j(\Gamma_j) - f_j(\Gamma_{\mathcal{M}}^*) \le \sum_{j=1}^{k(T)}(\Gamma_j - \Gamma_{\mathcal{M}}^*)^2(\gamma_j - \gamma_{j-1} - c_j) + \sum_{j=1}^{k(T)} \frac{1}{\gamma_j} h_j^2(\Gamma_j)$$

$$\text{since } \gamma_0 \triangleq 0, \gamma_j \ge jl_{\min}, \text{ and } |h_j(\Gamma_j)| \le G \triangleq n_{\max}\left(1 + \frac{l_{\max}}{l_{\min}}\right)$$

$$\le 0 + G^2 \sum_{j=1}^{k(T)} \frac{1}{jl_{\min}} \le \frac{G^2}{l_{\min}}\big[1 + \log(k(T))\big]$$

$$\text{since } k(T) \le \frac{T}{l_{\min}}$$

$$\le \frac{G^2}{l_{\min}}\left[1 + \log\left(\frac{T}{l_{\min}}\right)\right]. \tag{16}$$

**Step 4:** By Theorem 1, we have $\mathbb{E}[h_j(\Gamma_{\mathcal{M}}^*)] = 0$. Moreover, since $\Gamma_j$ depends only on the history $\mathcal{H}_{j-1}$, the sequence $\left\{ \sum_{j=1}^{k}(\Gamma_{\mathcal{M}}^* - \Gamma_j)\, h_j(\Gamma_{\mathcal{M}}^*)\right\}_{k=1}^{\infty}$ forms a martingale. Consequently,

$$\mathbb{E}\left[\sum_{j=1}^{k(T)}(\Gamma_{\mathcal{M}}^* - \Gamma_j)\, h_j(\Gamma_{\mathcal{M}}^*)\right] = 0. \tag{17}$$

Applying equation 17 together with equation 16 to equation 13, we obtain

$$\mathbb{E}\left[\left(\sum_{j=1}^{k(T)} l_{\max}|\Gamma_j - \Gamma_{\mathcal{M}}^*|\right)^2\right] \le T\left(\frac{l_{\max}}{l_{\min}}\right)^2 \frac{G^2}{l_{\min}}\left[1 + \log\left(\frac{T}{l_{\min}}\right)\right] \triangleq U_T.$$

Finally, since $\mathbb{E}[x] \le \sqrt{\mathbb{E}[x^2]}$ holds for any random variable x, we conclude

$$R_T^{\mathsf{C}} \le \mathbb{E}\left[\sum_{j=1}^{k(T)} l_{\max}|\Gamma_j - \Gamma_{\mathcal{M}}^*|\right] + l_{\max}\Gamma_{\max} \le \sqrt{U_T} + l_{\max}\Gamma_{\max}.$$

$\square$

# B    SUPPORTING LEMMAS FOR CONTEXTUAL BANDITS

In this section, we review relevant results from the existing contextual bandit literature and extend them for later regret analysis for COAF.

## B.1    CONCENTRATION PROPERTIES OF THE REGULARIZED LEAST SQUARES

In the standard contextual bandit setting, the learner selects a single arm at each round $k$. Let $\boldsymbol{x}_k$ denote the context and $y_k$ the noisy feedback from the chosen arm at round $k$. The noise in $y_k$ is conditionally sub-Gaussian, as stated in Assumption 3. The concentration inequality in this section ensures that the confidence set $\mathcal{F}_k \subseteq \mathcal{F}$ contains $\psi_*$ with high probability for all $k \in \mathbb{N}$. Consequently, the UCB of each arm is, with high probability, no smaller than its true mean reward.

### B.1.1    CONCENTRATION INEQUALITY FOR LINEAR REGRESSOR

In the linear bandit setting, the regularized least-squares estimator is defined as

$$\bar{\theta}_k = V_k^{-1}(\lambda) \sum_{j=1}^{k} \boldsymbol{x}_j y_j, \ V_k(\lambda) = \lambda I + \sum_{j=1}^{k} \boldsymbol{x}_j \boldsymbol{x}_j^\top.$$

**Lemma 5** ((Lattimore & Szepesvári, 2020, Theorem 20.5))**.** *Let $\delta \in (0, 1)$. Then, with probability at least $1 - \delta$, it holds that for all $k \in \mathbb{N}$,*

$$\left\|\bar{\theta}_k - \theta_*\right\|_{V_k(\lambda)} < \sqrt{\lambda}\,\|\theta_*\| + \sqrt{2\log\left(\frac{1}{\delta}\right) + \log\left(\frac{\det V_k(\lambda)}{\lambda^d}\right)}.$$

*Furthermore, if $\|\theta_*\| \leq m$, then $P(\exists k \in \mathbb{N} : \theta_* \notin \mathcal{C}_k) \leq \delta$ with*

$$\mathcal{C}_k = \left\{ \theta \in \mathbb{R}^d \,\middle|\, \left\| \theta - \bar{\theta}_{k-1} \right\|_{V_{k-1}(\lambda)} < m\sqrt{\lambda} + \sqrt{2\log\left(\tfrac{1}{\delta}\right) + \log\left(\tfrac{\det V_{k-1}(\lambda)}{\lambda^d}\right)} \right\}.$$

### B.1.2 Concentration Inequality for General Regressor

For a general regressor class $\mathcal{F}$, recall $N(\mathcal{F}, \alpha, \|\cdot\|_\infty)$ denotes its $\alpha$-covering number under the sup-norm $\|\cdot\|_\infty$. The regularized least-squares estimator is $\bar{\psi}_k \in \arg\min_{\psi \in \mathcal{F}} \sum_{j=1}^{k} \left( \psi(\boldsymbol{x}_j) - \mathrm{y}_j \right)^2$. The following result relates the concentration of $\bar{\psi}_k$ to the $\alpha$-covering number of $\mathcal{F}$.

**Lemma 6** ((Russo & Van Roy, 2013, Proposition 2)). *For any $\delta > 0$ and $\alpha > 0$, with probability at least $1 - 2\delta$, it holds for all $k \in \mathbb{N}$ that*

$$\psi_* \in \left\{ \psi \in \mathcal{F} \,\middle|\, \sum_{j=1}^{k-1} [\psi(\boldsymbol{x}_j) - \bar{\psi}_k(\boldsymbol{x}_j)]^2 \leq \tilde{\beta}(k, \mathcal{F}, \delta, \alpha) \right\},$$

*where $\tilde{\beta}(k, \mathcal{F}, \delta, \alpha) = 8\log\left( N(\mathcal{F}, \alpha, \|\cdot\|_\infty)/\delta \right) + 2\alpha k(8 + \sqrt{8\log(4k^2/\delta)})$.*

## B.2 Bounds for Cumulative Prediction Error

We report the upper bound on the cumulative prediction error of the estimated rewards. Combined with the results in Appendix B.1.1, this error is shown to be small with high probability.

### B.2.1 Cumulative Prediction Error with Linear Regressor Class

With a linear regressor class, the upper bound on the cumulative prediction error depends on the feature dimension $d$. The following result builds on the theory of self-normalized processes, and the version for standard contextual bandits appears in Lattimore & Szepesvári (2020, Lemma 19.4).

**Lemma 7.** *Let $\boldsymbol{V}_0 \in \mathbb{R}^{d \times d}$ be positive definite and $\boldsymbol{V}_k = \boldsymbol{V}_0 + \sum_{j=1}^{k} \sum_{i \in A_j} \left( 1 + \mathbf{x}_{j,i}^\top \boldsymbol{V}_{j-1}^{-1} \mathbf{x}_{j,i} \right)$. If $\|\mathbf{x}_{j,i}\| \leq L < \infty$ for all $i \in [\mathrm{n}_j]$ and $j \in \mathbb{N}$, then*

$$\sum_{j=1}^{k} \sum_{i \in A_j} \min\left( 1, \mathbf{x}_{j,i}^\top \boldsymbol{V}_{j-1}^{-1} \mathbf{x}_{j,i} \right) \leq 2\log\left( \frac{\det \boldsymbol{V}_k}{\det \boldsymbol{V}_0} \right) \leq 2d\log\left( \frac{\operatorname{trace} \boldsymbol{V}_0 + n_{\max} k L^2}{d \det(\boldsymbol{V}_0)^{1/d}} \right).$$

*Proof.* The matrix determinant lemma, which states that for any vector $x$ and positive definite $\boldsymbol{V}$,

$$\det(\boldsymbol{V} + \boldsymbol{x}\boldsymbol{x}^\top) = \det(\boldsymbol{V})(1 + \boldsymbol{x}^\top \boldsymbol{V}^{-1} \boldsymbol{x}).$$

Applying this iteratively for all $\mathbf{x}_{j,i}$ gives

$$\det(\boldsymbol{V}_k) = \det(\boldsymbol{V}_0) \prod_{j=1}^{k} \prod_{i \in A_j} \left( 1 + \mathbf{x}_{j,i}^\top \boldsymbol{V}_{j-1}^{-1} \mathbf{x}_{j,i} \right).$$

Taking logarithms and using $2\log(1+x) \geq \min(x, 1)$ for any $x \geq 0$, we have

$$2\log\left( \frac{\det \boldsymbol{V}_k}{\det \boldsymbol{V}_0} \right) = 2\sum_{j=1}^{k} \sum_{i \in A_j} \log\left( 1 + \mathbf{x}_{j,i}^\top \boldsymbol{V}_{j-1}^{-1} \mathbf{x}_{j,i} \right) \leq \sum_{j=1}^{k} \sum_{i \in A_j} \min\left( 1, \mathbf{x}_{j,i}^\top \boldsymbol{V}_{j-1}^{-1} \mathbf{x}_{j,i} \right).$$

Finally, using the trace-determinant inequality for positive definite matrices:

$$\det(\boldsymbol{V}_k) \leq \left( \frac{\operatorname{trace}(\boldsymbol{V}_k)}{d} \right)^d \leq \left( \frac{\operatorname{trace}(V_0) + n_{\max} k L^2}{d} \right)^d,$$

we obtain

$$\sum_{j=1}^{k} \sum_{i \in A_j} \mathbf{x}_{j,i}^\top \boldsymbol{V}_{j-1}^{-1} \mathbf{x}_{j,i} \leq 2\log\left( \frac{\det \boldsymbol{V}_k}{\det \boldsymbol{V}_0} \right) \leq 2d\log\left( \frac{\operatorname{trace} \boldsymbol{V}_0 + n_{\max} k L^2}{d \det(\boldsymbol{V}_0)^{1/d}} \right).$$

$\square$

**Lemma 8.** *Let $\beta_1, \beta_2, \dots$ be a nondecreasing sequence with $\beta_1 \geq 1$ and let*

$$\mathcal{C}_j = \left\{ \theta \in \mathbb{R}^d \;\middle|\; \left\| \theta - \bar{\theta}_{j-1} \right\|_{\boldsymbol{V}_{j-1}}^2 \leq \beta_j, \|\theta\| \leq 1 \right\}.$$

*For each $j \in \mathbb{N}$, take an arbituary pair $\theta_j, \theta_j' \in \mathcal{C}_j$. If $\|\mathbf{x}_{j,i}\| \leq 1$ for all $i \in [\mathrm{n}_j]$ and $j \in \mathbb{N}$, then*

$$\sum_{j=1}^{k} \sum_{i \in A_j} \langle \theta_j - \theta_j', \mathbf{x}_{j,i} \rangle^2 \leq 8d\beta_k \log \left( \frac{\mathrm{trace}\, \boldsymbol{V}_0 + n_{\max} k}{d \det(\boldsymbol{V}_0)^{1/d}} \right).$$

*Furthermore, if $\boldsymbol{V}_0 = \lambda I$,*

$$\sum_{j=1}^{k} \sum_{i \in A_j} \langle \theta_j - \theta_j', \mathbf{x}_{j,i} \rangle^2 \leq 8d\beta_k \log \left( \frac{d\lambda + n_{\max} k}{d\lambda} \right).$$

*Proof.* For a vectore $\boldsymbol{x} \in \mathbb{R}^d$ and a positive definite matrix $\boldsymbol{V} \in \mathbb{R}^{d \times d}$, recall $\|\boldsymbol{x}\|_{\boldsymbol{V}} = \sqrt{\boldsymbol{x}^\top \boldsymbol{V} \boldsymbol{x}}$. Since $\theta_j, \theta_j' \in \mathcal{C}_j$, for each $i \in [\mathrm{n}_j]$, we have

$$\langle \theta_j - \theta_j', \mathbf{x}_{j,i} \rangle \leq \|\mathbf{x}_{j,i}\|_{\boldsymbol{V}_{j-1}^{-1}} \left\| \theta_j - \theta_j' \right\|_{\boldsymbol{V}_{j-1}} \leq 2 \|\mathbf{x}_{j,i}\|_{\boldsymbol{V}_{j-1}^{-1}} \sqrt{\beta_j}.$$

With $\|\theta_j\|, \|\theta_j'\|$ and $\|\mathbf{x}_{j,i}\|$ all upper bounded by 1, we have $\left| \langle \theta_j - \theta_j', \mathbf{x}_{j,i} \rangle \right| \leq 2$. Combing this result with $\beta_k \geq \beta_j \geq 1$, we have

$$\langle \theta_j - \theta_j', \mathbf{x}_{j,i} \rangle \leq \min \left( 2, 2 \|\mathbf{x}_{j,i}\|_{V_{j-1}^{-1}} \sqrt{\beta_j} \right) \leq 2\sqrt{\beta_k} \min \left( 1, \|\mathbf{x}_{j,i}\|_{V_{j-1}^{-1}} \right).$$

We use the above inequality and apply Lemma 7 to get

$$\sum_{j=1}^{k} \sum_{i \in A_j} \langle \theta_j - \theta_j', \mathbf{x}_{j,i} \rangle^2 \leq 4\beta_k \sum_{j=1}^{k} \sum_{i \in A_j} \min \left( 1, \|\mathbf{x}_{j,i}\|_{\boldsymbol{V}_{j-1}^{-1}} \right) \leq 8d\beta_k \log \left( \frac{\mathrm{trace}\, \boldsymbol{V}_0 + n_{\max} k}{d \det(\boldsymbol{V}_0)^{1/d}} \right).$$

$\square$

### B.2.2 CUMULATIVE PREDICTION ERROR WITH GENERAL REGRESSOR CLASS

For an abstract regressor class $\mathcal{F}$, the upper bound of the cumulative prediction error depends on its eluder dimension $\dim_{\mathrm{E}}(\mathcal{F}, \epsilon)$. For any subset $\tilde{\mathcal{F}} \subseteq \mathcal{F}$, its width at a context $\boldsymbol{x}$ is defined as

$$w_{\tilde{\mathcal{F}}}(\boldsymbol{x}) \triangleq \sup_{\psi, \psi' \in \tilde{\mathcal{F}}} \psi(\boldsymbol{x}) - \psi'(\boldsymbol{x}).$$

**Single arm selection:** The following results apply to the standard contextual bandit setting, in which a single arm is selected from each decision set.

**Lemma 9** ((Russo & Van Roy, 2013, Proposition 3)). *Let $\boldsymbol{x}_1, \dots, \boldsymbol{x}_n \in \mathbb{R}^d$ be a sequence of features, and let $\beta_1, \dots, \beta_n$ be a nondecreasing sequence. For each $k \in [n]$ and an arbitrary $\psi_k' \in \mathcal{F}$, define $\mathcal{F}_k \triangleq \left\{ \psi \in \mathcal{F} \mid \sum_{j=1}^{k-1} [\psi(\boldsymbol{x}_j) - \psi'_k(\boldsymbol{x}_j)]^2 \leq \beta_k \right\}$. Then, for any $\epsilon > 0$,*

$$\sum_{k=1}^{n} \mathbf{1} \left\{ w_{\mathcal{F}_k'}(\boldsymbol{x}_k) > \epsilon \right\} \leq \left( \frac{4\beta_n}{\epsilon^2} + 1 \right) \dim_{\mathrm{E}}(\mathcal{F}, \epsilon).$$

**Lemma 10.** *Let $\boldsymbol{x}_1, \dots, \boldsymbol{x}_n \in \mathbb{R}^d$ be a sequence of features, and let $\beta_1, \dots, \beta_n$ be a nondecreasing sequence. For each $k \in [n]$ and an arbitrary $\psi_k' \in \mathcal{F}$, define $\mathcal{F}_k \triangleq \left\{ \psi \in \mathcal{F} \mid \sum_{i=1}^{k-1} [\psi(\boldsymbol{x}_i) - \psi_k(\boldsymbol{x}_i)]^2 \leq \beta_n \right\}$. Then the widths $w_{\mathcal{F}_1}(\boldsymbol{x}_1), \dots, w_{\mathcal{F}_n}(\boldsymbol{x}_n)$ satisfy*

$$\sum_{k=1}^{n} w_{\mathcal{F}_k}^2(\boldsymbol{x}_k) \leq 4 \dim_{\mathrm{E}} \left( \mathcal{F}, \tfrac{1}{n} \right) + 1 + 4\beta_n \dim_{\mathrm{E}} \left( \mathcal{F}, \tfrac{1}{n} \right) (1 + \log n).$$

*Proof.* For the ease of notation, let $w_{\mathcal{F}_k}(\boldsymbol{x}_k) = w_k$. We rearrange the sequence $w_1, \ldots, w_n$ by defining a sequence $k_1, \ldots, k_n$ such that $w_{k_1} \geq w_{k_2} \geq \ldots \geq w_{k_n}$. For any $w_{k_{i+1}} > \frac{1}{n}$, there are at most $i$ values in $w_1, \ldots, w_n$ that is greater $w_{k_{i+1}}$, and we apply Lemma 9 to get

$$i \leq \sum_{k=1}^n \mathbf{1}\left\{w_{\mathcal{F}_k}(\boldsymbol{x}_k) > w_{k_{i+1}}\right\}$$

$$\leq \left(\frac{4\beta_n}{w_{k_{i+1}}^2} + 1\right) \dim_{\mathrm{E}}(\mathcal{F}, w_{k_{i+1}}) \leq \left(\frac{4\beta_n}{w_{k_{i+1}}^2} + 1\right) \dim_{\mathrm{E}}\left(\mathcal{F}, \tfrac{1}{n}\right),$$

where the last inequality is due to that $\dim_{\mathrm{E}}(\mathcal{F}, \epsilon)$ is a nonincreasing function of $\epsilon$. Thus, for any $i > m = \dim_{\mathrm{E}}(\mathcal{F}, \frac{1}{n})$, we rearrange the above inequality to get

$$w_{k_{i+1}}^2 \leq \frac{4\beta_n m}{i - m}, \quad \text{if } w_{k_{i+1}} > \frac{1}{n}.$$

Moreover, since the range of each $\psi \in \mathcal{F}$ is contained in $[-1, 1]$, $w_k \leq 2$ for all $k \in [n]$. Thus,

$$\sum_{k=1}^n w_k^2 = \sum_{i=1}^n w_{k_i}^2 \leq 4m + \sum_{i=m+1}^n w_{k_i}^2 \mathbf{1}\left\{w_{k_i} \leq \frac{1}{n}\right\} + \sum_{i=m+1}^n w_{k_i}^2 \mathbf{1}\left\{w_{k_i} > \frac{1}{n}\right\}$$

$$< 4m + \frac{1}{n} + \sum_{i=m+1}^n \frac{4\beta_n m}{i - m} \leq 4m + 1 + 4\beta_n m(1 + \log n).$$

$\square$

**Multiple arm selection:** When multiple arms are selected in each round, we define

$$\mathcal{F}_k \triangleq \left\{\psi \in \mathcal{F} \;\Big|\; \sum_{j=1}^{k-1} \sum_{i \in A_j} [\psi(\mathbf{x}_{j,i}) - \psi'_k(\mathbf{x}_{j,i})]^2 \leq \beta_k\right\}.$$

To see the difference, we show the feature sequence below:

$$\underbrace{\mathbf{x}_{1,1}, \ldots, \mathbf{x}_{1,|A_1|}, \ldots, \mathbf{x}_{j-1,1}, \ldots, \mathbf{x}_{j-1,|A_{j-1}|}}_{(a)}, \underbrace{\mathbf{x}_{j,1}, \ldots, \mathbf{x}_{j,i-1}}_{(b)}, \underset{\uparrow}{\mathbf{x}_{j,i}}, \ldots, \mathbf{x}_{n,1}, \ldots, \mathbf{x}_{j-1,|A_n|}.$$

When evaluating the width at $\mathbf{x}_{j,i}$, the set $\mathcal{F}_j$ is constructed by imposing constraints only on the features in $(a)$; the features in $(b)$ are not incorporated. For each $i \in A_j$ and $j \in \mathbb{N}$, define

$$\mathcal{F}_{j,i}^{\mathrm{m}} \triangleq \left\{\psi \in \mathcal{F} \;\Big|\; \sum_{\nu=1}^{j-1} \sum_{\iota \in A_\nu} [\psi(\mathbf{x}_{\nu,\iota}) - \psi'_k(\mathbf{x}_{\nu,\iota})]^2 + \sum_{\iota=1}^{i-1} [\psi(\mathbf{x}_{j,\iota}) - \psi'_k(\mathbf{x}_{j,\iota})]^2 \leq \beta_j + n_{\max}\right\}.$$

With Lemma 9, we obtain

$$\sum_{j=1}^n \sum_{i \in A_j} \mathbf{1}\left\{w_{\mathcal{F}_{j,i}^{\mathrm{m}}}(\mathbf{x}_{j,i}) > \epsilon\right\} \leq \left[\frac{4(\beta_n + 4n_{\max})}{\epsilon^2} + 1\right] \dim_{\mathrm{E}}(\mathcal{F}, \epsilon).$$

Since $\sum_{\iota=1}^{i-1} [\psi(\mathbf{x}_{j,\iota}) - \psi'_j(\mathbf{x}_{j,\iota})]^2 \leq 4n_{\max}$, we also have $\mathcal{F}_{j,i}^{\mathrm{m}} \supseteq \mathcal{F}_j$. We get the following corollary.

**Corollary 11.** *let $\beta_1, \ldots, \beta_n$ be a nondecrasing sequence, then for any $\epsilon > 0$,*

$$\sum_{j=1}^n \sum_{i \in A_j} \mathbf{1}\left\{w_{\mathcal{F}_j}(\mathbf{x}_{j,i}) > \epsilon\right\} \leq \left[\frac{4(\beta_n + 4n_{\max})}{\epsilon^2} + 1\right] \dim_{\mathrm{E}}(\mathcal{F}, \epsilon).$$

Applying the same steps that lead from Lemma 9 to Lemma 10, we derive the following corollary from Corollary 11.

**Corollary 12.** *Let $\beta_1, \ldots, \beta_n$ be a nondecreasing sequence. For each $k \in [n]$, let*

$$\mathcal{F}_k \triangleq \left\{\psi \in \mathcal{F} \;\Big|\; \textstyle\sum_{j=1}^{k-1} \sum_{i \in A_j} [\psi(\mathbf{x}_{j,i}) - \bar{\psi}_k(\mathbf{x}_{j,i})]^2 \leq \beta_k\right\},$$

*and let $\mathrm{s}_n = \sum_{j=1}^n |A_j|$, then*

$$\sum_{j=1}^n \sum_{i \in A_j} w_{\mathcal{F}_j}^2(\mathbf{x}_{j,i}) \leq 4 \dim_{\mathrm{E}}\left(\mathcal{F}, \frac{1}{\mathrm{s}_n}\right) + 1 + (4\beta_n + 4n_{\max}) \dim_{\mathrm{E}}\left(\mathcal{F}, \frac{1}{\mathrm{s}_n}\right) (1 + \log(\mathrm{s}_n)).$$

## C    REGRET ANALYSIS FOR COAF

The following result provides a decomposition of the regret for COAF, isolating the components that arise from learning the mean reward function using online feedback.

**Lemma 13.** *If $\psi_* \in \mathcal{F}_j$ for all $j \in \mathbb{N}$, the regret for COAF satisfies*

$$R_T^C \leq \sqrt{\underbrace{\frac{U_T}{\xi} + \frac{T}{1-\xi}\left(\frac{l_{\max}}{l_{\min}}\right)^3 \mathbb{E}\left[\sum_{j=1}^{k(T)}\sum_{i\in A_j}(\hat{\mu}_{j,i}-\mu_{j,i})^2\right]}_{(a)}}$$

$$+ \underbrace{\sqrt{\frac{Tn_{\max}}{l_{\min}}\mathbb{E}\left[\sum_{j=1}^{k(T)}\sum_{i\in A_j}(\hat{\mu}_{j,i}-\mu_{j,i})^2\right]}}_{(b)} + l_{\max}\Gamma_{\max},$$

*where $U_T \triangleq \frac{T}{l_{\min}}\left(\frac{l_{\max}n_{\max}}{l_{\min}}\right)^2\left(1+\frac{l_{\max}}{l_{\min}}\right)^2\left[1+\log\left(\frac{T}{l_{\min}}\right)\right].$*

Components (a) and (b) arise from the use of UCB estimates in place of the true mean rewards. The remaining terms coincide with those in the oracle case, as established in Lemma 2.

*Proof.* In equation 7, we have shown

$$R_T^C = \mathbb{E}\left[\sum_{j=1}^{k(T)} g(\Gamma_{\mathcal{M}}^*, A_j, 1_j, \boldsymbol{\mu}_j)\right] + l_{\max}\Gamma_{\max}. \tag{18}$$

At each round $j$, with estimated arm rewards $\hat{\boldsymbol{\mu}}_j$, COAF selects a subset of arms

$$A_j \in \underset{A\in\mathbb{A}_j}{\arg\min}\, g(\Gamma_j, A, 1_j, \hat{\boldsymbol{\mu}}_j). \tag{19}$$

Correspondingly, the optimal arm selection is defined as

$$A_j^* \in \underset{A\in\mathbb{A}_j}{\arg\min}\, g(\Gamma_{\mathcal{M}}^*, A, 1_j, \boldsymbol{\mu}_j).$$

**Step 1:** To bound $g(\Gamma_{\mathcal{M}}^*, A_j, 1_j, \boldsymbol{\mu}_j)$, we consider two seperate cases: $\Gamma_j < \Gamma_{\mathcal{M}}^*$ and $\Gamma_j \geq \Gamma_{\mathcal{M}}^*$.

*Case 1:* If $\Gamma_j < \Gamma_{\mathcal{M}}^*$, then

$$g(\Gamma_{\mathcal{M}}^*, A_j, 1_j, \boldsymbol{\mu}_j) = 1_j(A_j)\Gamma_{\mathcal{M}}^* - \sum_{i\in A_j}\mu_{j,i}$$

$$= 1_j(A_j)(\Gamma_{\mathcal{M}}^* - \Gamma_j) + 1_j(A_j)\Gamma_j - \sum_{i\in A_j}\hat{\mu}_{j,i} + \sum_{i\in A_j}(\hat{\mu}_{j,i} - \mu_{j,i})$$

according to equation 19

$$\leq 1_j(A_j)(\Gamma_{\mathcal{M}}^* - \Gamma_j) + 1_j(A_j^*)\Gamma_j - \sum_{i\in A_j^*}\hat{\mu}_{j,i} + \sum_{i\in A_j}(\hat{\mu}_{j,i} - \mu_{j,i}).$$

Moreover, if $\psi_* \in \mathcal{F}_j$, then for each arm $i \in [n_j]$, the UCB estimate satisfies

$$\hat{\mu}_{j,i} = \max_{\psi\in\mathcal{F}_j}\psi(\mathbf{x}_{j,i}) \geq \mu_{j,i}.$$

Hence, we obtain

$$g(\Gamma_{\mathcal{M}}^*, A_j, 1_j, \boldsymbol{\mu}_j) \leq 1_j(A_j)(\Gamma_{\mathcal{M}}^* - \Gamma_j) + 1_j(A_j^*)\Gamma_{\mathcal{M}}^* - \sum_{i\in A_j^*}\mu_{j,i} + \sum_{i\in A_j}(\hat{\mu}_{j,i} - \mu_{j,i})$$

$$= 1_j(A_j)(\Gamma_{\mathcal{M}}^* - \Gamma_j) + g(\Gamma_{\mathcal{M}}^*, A_j^*, 1_j, \boldsymbol{\mu}_j) + \sum_{i\in A_j}(\hat{\mu}_{j,i} - \mu_{j,i}).$$

*Case 2:* If $\Gamma_j \geq \Gamma_{\mathcal{M}}^*$, we get

$$g(\Gamma_{\mathcal{M}}^*, A_j, 1_j, \boldsymbol{\mu}_j) \leq 1_j(A_j)\Gamma_{\mathcal{M}}^* - \sum_{i \in A_j} \mu_{j,i}$$

$$\leq 1_j(A_j)\Gamma_j - \sum_{i \in A_j} \hat{\mu}_{j,i} + \sum_{i \in A_j} (\hat{\mu}_{j,i} - \mu_{j,i})$$

according to equation 19

$$\leq 1_j(A_j^*)\Gamma_j - \sum_{i \in A_j^*} \hat{\mu}_{j,i} + \sum_{i \in A_j} (\hat{\mu}_{j,i} - \mu_{j,i})$$

since as $\hat{\mu}_{j,i} \geq \mu_{j,i}$ for all $i$

$$\leq 1_j(A_j^*)(\Gamma_j - \Gamma_{\mathcal{M}}^*) + 1_j(A_j^*)\Gamma_{\mathcal{M}}^* - \sum_{i \in A_j^*} \mu_{j,i} + \sum_{i \in A_j} (\hat{\mu}_{j,i} - \mu_{j,i})$$

$$= 1_j(A_j^*)(\Gamma_j - \Gamma_{\mathcal{M}}^*) + g(\Gamma_{\mathcal{M}}^*, A_j^*, 1_j, \boldsymbol{\mu}_j) + \sum_{i \in A_j} (\hat{\mu}_{j,i} - \mu_{j,i}).$$

With maximum latency $l_{\max}$, we combine both cases together to get

$$g(\Gamma_{\mathcal{M}}^*, A_j, 1_j, \boldsymbol{\mu}_j) \leq l_{\max} |\Gamma_j - \Gamma_{\mathcal{M}}^*| + g(\Gamma_{\mathcal{M}}^*, A_j^*, 1_j, \boldsymbol{\mu}_j) + \sum_{i \in A_j} (\hat{\mu}_{j,i} - \mu_{j,i}).$$

It follows from Theorem 1 that $\mathbb{E}[g(\Gamma_{\mathcal{M}}^*, A_j^*, 1_j, \boldsymbol{\mu}_j)] = 0$. We apply this result to equation 18 to get

$$R_T^{\mathrm{C}} \leq \mathbb{E}\left[\sum_{j=1}^{\mathrm{k}(T)} l_{\max} |\Gamma_j - \Gamma_{\mathcal{M}}^*| + \sum_{i \in A_j} (\hat{\mu}_{j,i} - \mu_{j,i})\right] + l_{\max}\Gamma_{\max}. \tag{20}$$

**Step 2:** Recall $h_j(\Gamma)$ defined in equation 10. We define its counterpart with UCB estimates

$$\hat{h}_j(\Gamma) = \min_{A \in \mathbb{A}_j} g(\Gamma, A, 1_j, \hat{\boldsymbol{\mu}}_j).$$

Then we have the following

$$h_j(\Gamma_j) = \min_{A \in \mathbb{A}_j} \left[ 1_j(A)\Gamma - \sum_{i \in A} \mu_i \right]$$

$$\leq 1_j(A_j)\Gamma_j - \sum_{i \in A_j} \mu_{j,i} = 1_j(A_j)\Gamma_j - \sum_{i \in A_j} \hat{\mu}_{j,i} + \sum_{i \in A_j} (\hat{\mu}_{j,i} - \mu_{j,i})$$

$$= \hat{h}_j(\Gamma_j) + \sum_{i \in A_j} (\hat{\mu}_{j,i} - \mu_{j,i}). \tag{21}$$

Let $f_j(\Gamma)$ be such that $f_j'(\Gamma) = h_j(\Gamma)$. In step 2 of the proof for Lemma 2, we have shown

$$\left(\sum_{j=1}^{\mathrm{k}(T)} l_{\max} |\Gamma_j - \Gamma_{\mathcal{M}}^*|\right)^2 \leq 2T\left(\frac{l_{\max}}{l_{\min}}\right)^2 \sum_{j=1}^{\mathrm{k}(T)} \left[ f_j(\Gamma_j) - f_j(\Gamma_{\mathcal{M}}^*) + (\Gamma_{\mathcal{M}}^* - \Gamma_j)h_j(\Gamma_{\mathcal{M}}^*) \right]. \tag{22}$$

In later steps of the proof, we use $\hat{h}_j$ to bound the above term.

**Step 3:** We continue to give an upper bound on $\sum_{j=1}^{\mathrm{k}(T)} f_j(\Gamma_j) - f_j(\Gamma_{\mathcal{M}}^*)$. Since $f_j$ is strongly convex with parameter $c_j$,

$$2\big[f_j(\Gamma_j) - f_j(\Gamma_{\mathcal{M}}^*)\big]$$

$$\leq 2h_j(\Gamma_j)(\Gamma_j - \Gamma_{\mathcal{M}}^*) - c_j(\Gamma_j - \Gamma_{\mathcal{M}}^*)^2$$

$$= 2\hat{h}_j(\Gamma_j)(\Gamma_j - \Gamma_{\mathcal{M}}^*) + 2\big[h_j(\Gamma_j) - \hat{h}_j(\Gamma_j)\big](\Gamma_j - \Gamma_{\mathcal{M}}^*) - c_j(\Gamma_j - \Gamma_{\mathcal{M}}^*)^2$$

since $2\big[h_j(\Gamma_j) - \hat{h}_j(\Gamma_j)\big](\Gamma_j - \Gamma_{\mathcal{M}}^*) \leq \frac{1}{a_j}\big[h_j(\Gamma_j) - \hat{h}_j(\Gamma_j)\big]^2 + a_j(\Gamma_j - \Gamma_{\mathcal{M}}^*)^2$

$$\leq 2\hat{h}_j(\Gamma_j)(\Gamma_j - \Gamma_{\mathcal{M}}^*) + \frac{1}{a_j}\big[h_j(\Gamma_j) - \hat{h}_j(\Gamma_j)\big]^2 + (a_j - c_j)(\Gamma_j - \Gamma_{\mathcal{M}}^*)^2, \tag{23}$$

for some $a_j > 0$ to be chosen in later steps. Using equation 21 and applying the Cauchy–Schwarz inequality, we obtain

$$
\begin{aligned}
\left[h_j(\Gamma_j) - \hat{h}_j(\Gamma_j)\right]^2 &\leq \left[\sum_{i \in A_j} (\hat{\mu}_{j,i} - \mu_{j,i})\right]^2 \\
&\leq |A_j| \sum_{i \in A_j} (\hat{\mu}_{j,i} - \mu_{j,i})^2 \leq l_{\max} \sum_{i \in A_j} (\hat{\mu}_{j,i} - \mu_{j,i})^2.
\end{aligned} \tag{24}
$$

In COAF as presented in Algorithm 1, $\Gamma_{j+1} = \Pi_{[\Gamma_{\min}, \Gamma_{\max}]} \left[\Gamma_j - \frac{1}{\xi\gamma_j} g(\Gamma_j, A_j, 1_j, \hat{\boldsymbol{\mu}}_j)\right]$. Then we apply the fact that $g(\Gamma_j, A_j, 1_j, \hat{\boldsymbol{\mu}}_j) = \hat{h}_j(\Gamma_j)$ to get

$$
\begin{aligned}
(\Gamma_{j+1} - \Gamma_{\mathcal{M}}^*)^2 &= \left[\Pi_{[\Gamma_{\min}, \Gamma_{\max}]}\left(\Gamma_j - \frac{1}{\xi\gamma_j}\hat{h}_j(\Gamma_j)\right) - \Gamma_{\mathcal{M}}^*\right]^2 \leq \left[\Gamma_j - \frac{1}{\xi\gamma_j}\hat{h}_j(\Gamma_j) - \Gamma_{\mathcal{M}}^*\right]^2 \\
&= (\Gamma_j - \Gamma_{\mathcal{M}}^*)^2 + \frac{1}{\xi^2\gamma_j^2}\hat{h}_j^2(\Gamma_j) - \frac{2}{\xi\gamma_j}\hat{h}_j(\Gamma_j)(\Gamma_j - \Gamma_{\mathcal{M}}^*).
\end{aligned}
$$

Rearranging the above equation, we get

$$
2\hat{h}_j(\Gamma_j)(\Gamma_j - \Gamma_{\mathcal{M}}^*) = \xi\gamma_j(\Gamma_j - \Gamma_{\mathcal{M}}^*)^2 - \xi\gamma_j(\Gamma_{j+1} - \Gamma_{\mathcal{M}}^*)^2 + \frac{1}{\xi\gamma_j}\hat{h}_j^2(\Gamma_j). \tag{25}
$$

Substituting equation 24 and equation 25 into equation 23, we compute the following summation:

$$
2\sum_{j=1}^{k(T)} f_j(\Gamma_j) - f_j(\Gamma_{\mathcal{M}}^*)
$$

$$
\leq \sum_{j=1}^{k(T)} (\Gamma_j - \Gamma_{\mathcal{M}}^*)^2(\xi\gamma_j - \xi\gamma_{j-1} + a_j - c_j) + \sum_{j=1}^{k(T)} \frac{1}{\xi\gamma_j}\hat{h}_j^2(\Gamma_j) + \frac{l_{\max}}{a_j}\sum_{j=1}^{k(T)}\sum_{i \in A_j}(\hat{\mu}_{j,i} - \mu_{j,i})^2
$$

since $\frac{1}{\gamma_0} \triangleq 0$, $\left|\hat{h}_j(\Gamma_j)\right| \leq G \triangleq n_{\max}\left(1 + \frac{l_{\max}}{l_{\min}}\right)$, and we select $a_j = (1 - \xi)c_j$

$$
= 0 + G^2 \sum_{j=1}^{k(T)} \frac{1}{\xi\gamma_j} + \frac{l_{\max}}{a_j}\sum_{j=1}^{k(T)}\sum_{i \in A_j}(\hat{\mu}_{j,i} - \mu_{j,i})^2
$$

since $c_j \geq l_{\min}$

$$
\leq 0 + \frac{G^2}{\xi}\sum_{j=1}^{k(T)} \frac{1}{jl_{\min}} + \frac{l_{\max}}{(1-\xi)l_{\min}}\sum_{j=1}^{k(T)}\sum_{i \in A_j}(\hat{\mu}_{j,i} - \mu_{j,i})^2
$$

$$
\leq \frac{G^2}{\xi l_{\min}}\left[1 + \log(k(T))\right] + \frac{l_{\max}}{(1-\xi)l_{\min}}\sum_{j=1}^{k(T)}\sum_{i \in A_j}(\hat{\mu}_{j,i} - \mu_{j,i})^2
$$

since $k(T) \leq \frac{T}{l_{\min}}$

$$
\leq \frac{G^2}{\xi l_{\min}}\left[1 + \log\left(\frac{T}{l_{\min}}\right)\right] + \frac{l_{\max}}{(1-\xi)l_{\min}}\sum_{j=1}^{k(T)}\sum_{i \in A_j}(\hat{\mu}_{j,i} - \mu_{j,i})^2. \tag{26}
$$

**Step 4:** In equation 17, we have shown

$$
\mathbb{E}\left[\sum_{j=1}^{k(T)}(\Gamma_{\mathcal{M}}^* - \Gamma_j)\,h_j(\Gamma_{\mathcal{M}}^*)\right] = 0.
$$

Applying it together with equation 26 to equation 22, we obtain

$$
\mathbb{E}\left[\left(\sum_{j=1}^{k(T)} l_{\max}|\Gamma_j - \Gamma_{\mathcal{M}}^*|\right)^2\right] \leq \frac{U_T}{\xi} + \frac{T}{1-\xi}\left(\frac{l_{\max}}{l_{\min}}\right)^3 \mathbb{E}\left[\sum_{j=1}^{k(T)}\sum_{i \in A_j}(\hat{\mu}_{j,i} - \mu_{j,i})^2\right].
$$

With this result, and using the fact that $\mathbb{E}[x] \leq \sqrt{\mathbb{E}[x^2]}$ for any random variable x, it follows from equation 20 that

$$R_T^{\mathrm{C}} \leq \sqrt{\frac{U_T}{\xi} + \frac{T}{1-\xi}\left(\frac{l_{\max}}{l_{\min}}\right)^3 \mathbb{E}\left[\sum_{j=1}^{\mathrm{k}(T)} \sum_{i \in A_j} (\hat{\mu}_{j,i} - \mu_{j,i})^2\right]}$$

$$+ \mathbb{E}\left[\sum_{j=1}^{\mathrm{k}(T)} \sum_{i \in A_j} (\hat{\mu}_{j,i} - \mu_{j,i})\right] + l_{\max}\Gamma_{\max}. \tag{27}$$

In addition, Cauchy–Schwarz inequality gives

$$\left[\sum_{j=1}^{\mathrm{k}(T)} \sum_{i \in A_j} (\hat{\mu}_{j,i} - \mu_{j,i})\right]^2 \leq \sum_{j=1}^{\mathrm{k}(T)} |A_j| \times \sum_{j=1}^{\mathrm{k}(T)} \sum_{i \in A_j} (\hat{\mu}_{j,i} - \mu_{j,i})^2 \leq \frac{T n_{\max}}{l_{\min}} \sum_{j=1}^{\mathrm{k}(T)} \sum_{i \in A_j} (\hat{\mu}_{j,i} - \mu_{j,i})^2.$$

Applying $\mathbb{E}[x] \leq \sqrt{\mathbb{E}[x^2]}$ to the second expectation in equation 27, and then combining it with the above inequality, we conclude the proof. $\qquad\square$

With Lemma 13 in place, we leverage the theoretical results for standard contextual bandits in Appendix B to analyze the regret of COAF. For some $\delta > 0$, we define event

$$\mathcal{E} \triangleq \{\forall k \in \mathbb{N} : \psi_* \in \mathcal{F}_k\}.$$

## C.1 Regret Upper Bound with Linear Regressor Class

*Proof of Theorem 3.* For a linear mean reward function $\psi_* \in \mathcal{F}_1$, the event $\mathcal{E}$ can also be written as

$$\mathcal{E} = \{\forall j \in \mathbb{N} : \theta_* \in \mathcal{C}_j\},$$

where

$$\mathcal{C}_j = \left\{\theta \in \mathbb{R}^d \,\middle|\, \|\theta - \bar{\theta}_{j-1}\|_{\mathbf{V}_{j-1}(\lambda)}^2 < \beta(\mathbf{s}_{j-1}, \delta), \|\theta\| \leq 1\right\}.$$

By Lemma 7, we also have $P(\neg\mathcal{E}) \leq \delta$, using the fact that $\det V_n(\lambda) \leq (\lambda + n/d)^d$.

For each $\mathbf{x}_{j,i}$, let $\hat{\theta}_{j,i} \in \arg\max_{\theta \in \mathcal{C}_j} \langle\theta, \mathbf{x}_{j,i}\rangle$, i.e., $\hat{\mu}_{j,i} = \langle\hat{\theta}_{j,i}, \mathbf{x}_{j,i}\rangle$. Since $\delta \in (0, 1/\sqrt{e}\,]$ ensures $\beta(0, \delta) \geq 1$, if the event $\mathcal{E}$ occurs, we can apply Lemma 8 to obtain

$$\sum_{j=1}^{\mathrm{k}(T)} \sum_{i \in A_j} (\hat{\mu}_{j,i} - \mu_{j,i})^2 = \sum_{j=1}^{\mathrm{k}(T)} \sum_{i \in A_j} \langle\hat{\theta}_{j,i} - \theta_*, \mathbf{x}_{j,i}\rangle^2 \leq 8d\beta(\mathbf{s}_{\mathrm{k}(T)}, \delta) \log\left(\frac{d\lambda + n_{\max}\mathrm{k}(T)}{d\lambda}\right),$$

where $\mathbf{s}_k = \sum_{j=1}^k |A_j|$. Since $\mathrm{k}(T) \leq T/l_{\min}$ and $\mathbf{s}_{\mathrm{k}(T)} \leq T n_{\max}/l_{\min}$, we also have

$$\sum_{j=1}^{\mathrm{k}(T)} \sum_{i \in A_j} (\hat{\mu}_{j,i} - \mu_{j,i})^2 \leq \frac{W_T(\delta)}{T}. \tag{28}$$

If $\mathcal{E}$ occurs, we apply equation 28 to Lemma 13 to get

$$R_T^{\mathrm{C}} \leq \sqrt{\frac{U_T}{\xi} + \frac{1}{1-\xi}\left(\frac{l_{\max}}{l_{\min}}\right)^3 W_T(\delta)} + \sqrt{\frac{n_{\max}}{l_{\min}} W_T(\delta)} + l_{\max}\Gamma_{\max},$$

which holds with probability at least $1 - \delta$.

$\qquad\square$

## C.2 Regret Upper Bound for General Regressor Class

*Proof of Theorem 4.* Using the concentration property of $\bar{\psi}_k$ from Lemma 6, we have $P(\neg\mathcal{E}) \leq 2\delta$.

For each $\mathbf{x}_{j,i}$, let $\hat{\psi}_{j,i} \in \arg\max_{\psi \in \mathcal{F}_j}(\mathbf{x}_{j,i})$, i.e., $\hat{\mu}_{j,i} = \hat{\psi}_{j,i}(\mathbf{x}_{j,i})$. Conditioned on the event $\mathcal{E}$, we also have $\psi_* \in \mathcal{F}_j$ for any $j \in \mathbb{N}$ and $i \in [\mathrm{n}_j]$. Thus,

$$\sum_{j=1}^{\mathrm{k}(T)} \sum_{i \in A_j} (\hat{\mu}_{j,i} - \mu_{j,i})^2 = \sum_{j=1}^{\mathrm{k}(T)} \sum_{i \in A_j} \left[\hat{\psi}_{j,i}(\mathbf{x}_{j,i}) - \psi_*(\mathbf{x}_{j,i})\right]^2 \leq \sum_{j=1}^{\mathrm{k}(T)} \sum_{i \in A_j} w_{\mathcal{F}_j}^2(\mathbf{x}_{j,i}).$$

Then we apply Corollary 12 to get

$$
\sum_{j=1}^{\mathrm{k}(T)} \sum_{i \in A_j} w_{\mathcal{F}_j}^2(\mathbf{x}_{j,i})
$$

$$
\leq 4 \dim_{\mathrm{E}}\left(\mathcal{F}, \frac{1}{\mathrm{s}_{\mathrm{k}(T)}}\right) + 1 + 4\left[\tilde{\beta}(\mathrm{k}(T), \mathcal{F}, \delta, \alpha) + 4n_{\max}\right] \dim_{\mathrm{E}}\left(\mathcal{F}, \frac{1}{\mathrm{s}_{\mathrm{k}(T)}}\right)\left(1 + \log(\mathrm{s}_{\mathrm{k}(T)})\right)
$$

since $\mathrm{k}(T) \leq T/l_{\min}$ and $\mathrm{s}_{\mathrm{k}(T)} \leq Tn_{\max}/l_{\min}$

$$
\leq 4 \dim_{\mathrm{E}}\left(\mathcal{F}, \frac{l_{\min}}{Tn_{\max}}\right) + 1 + 4\left[\tilde{\beta}\left(\frac{T}{n_{\min}}, \mathcal{F}, \delta, \alpha\right) + 4n_{\max}\right] \dim_{\mathrm{E}}\left(\mathcal{F}, \frac{l_{\min}}{Tn_{\max}}\right)\left(1 + \log \frac{Tn_{\max}}{l_{\min}}\right),
\tag{29}
$$

where $\mathrm{s}_k = \sum_{j=1}^{k} |A_j|$. We then apply Eq. (29) to Lemma 13 to get

$$
R_T^{\mathrm{C}} \leq \sqrt{\frac{U_T}{\xi} + \frac{1}{1-\xi}\left(\frac{l_{\max}}{l_{\min}}\right)^3 \tilde{W}_T(\delta)} + \sqrt{\frac{n_{\max}}{l_{\min}} \tilde{W}_T(\delta)} + \frac{n_{\max} l_{\max}}{l_{\min}},
$$

which holds with probability at least $1 - 2\delta$. $\qquad\square$

