# OpenReview forum: "Latency-Aware Contextual Bandit: Application to Cryo-EM Data Collection"
_ICLR.cc/2026/Conference — Submitted to ICLR 2026_

### Official Review · Reviewer_rV3c · 2025-10-25

**Soundness:** 3
**Presentation:** 3
**Contribution:** 2
**Rating:** 6
**Confidence:** 2

**Summary:**

In this paper, the authors studied a latency-aware contextual bandit framework that extends standard contextual bandits by incorporating the action delays and formulates it as a special case of a semi-Markov decision process. The authors proposed the Contextual Online Arm Filtering (COAF) algorithm, which combines stochastic approximation and UCB exploration to balance reward and latency. The authors provided theoretical analysis of their algorithm, proving sublinear regret bounds. Finally, they conducted numerical experiments on MovieLens and cryo-EM data to demonstrate that COAF outperforms baselines and improves data collection efficiency.

**Strengths:**

- The proposed latency-aware model generalizes contextual and combinatorial bandits by explicitly accounting for temporal costs. This is a novel problem in the bandits literature.
- The theoretical analysis appears sound and comprehensive, though I have not checked every proof in detail).
- I also appreciate the discussion of the application to cryo-EM data collection, which highlights the real-world relevance of the framework. Modeling microscope exposure and movement as latency is a strong and realistic motivation that grounds the theoretical development.

**Weaknesses:**

- While the latency-aware formulation is novel, COAF primarily builds on existing tools such as UCB and stochastic approximation. The conceptual combination is interesting but may feel incremental without deeper theoretical or algorithmic insights. Could you clarify whether the current results provide any new algorithmic intuition or theoretical implications for the broader bandit literature?

- The numerical experiments, though illustrative, are relatively small-scale, so the insights they provide are somewhat limited. The cryo-EM evaluation appears to use simulated data, with experiments mainly comparing COAF to human microscopists. Could you offer some more comprehensive analysis here? E.g., an ablation study examining how COAF’s performance changes under different latency distributions. Similarly, the MovieLens experiments feel limited in scope, particularly in the choice of baselines. It would be informative to also compare against a number of standard contextual bandit algorithms.

- Finally, it would be valuable for the authors to discuss additional application domains where the proposed latency-aware bandit framework could be beneficial, beyond the cryo-EM setting.

**Questions:**

See weaknesses.

---

> ### Author Response · Authors · 2025-11-29
> **Response to paper review**
>
> We would like to thank the reviewer for the detailed feedback. We would like to address the weakness here.
> - In bandit literature, there exist Mortal bandits, thresholding bandits, and combinatorial bandits. Different from all those setups, our new problem formulation indicates that selecting which arm and how many arms from a decision set, in many situations, should depend on the time cost of an action. The semi-Markov decision process (SMDP) could address the action latency. Since MAB is a special case of learning an MDP, our problem can be viewed as a special case of learning an SMDP. With some assumptions, we show that strong performance guarantees can be provided.
> - The cryo-EM data comes from microscope logs, so all the latencies are fixed. We just analyzed how much improvement we could provide with our algorithm, without human intervention. We were trying to show the limitations of existing algorithms in addressing our problem in the MovieLens experiments. The existing bandit algorithm will result in linear regret, as shown in Fig. 2 of the revised manuscript.
> - Our problem intends to optimize resource allocation in scenarios where both context gathering and service delivery incur significant time costs. Contexts, such as questionnaire feedback, credit reports, and medical records, often require substantial collection effort. Furthermore, high-value services—including financial, legal, and consulting—demand extensive human labor. Therefore, customer selection must strategically balance these time-based costs against potential revenue.

---

### Official Review · Reviewer_iCp9 · 2025-11-01

**Soundness:** 3
**Presentation:** 3
**Contribution:** 2
**Rating:** 4
**Confidence:** 4

**Summary:**

The authors study the contextual MAB problem where each action incurs a context dependent time cost and the goal is to maximize the reward per unit time. They develop new algorithm called COAF that jointly learns the reward model with UCB style confidence band and also learns the optimal average reward. They also showcase the performance of the algorithm with theoretical guarantees with regret bound in several regimes.  The theoretical work is supported with experiments conducted on two real world data from different domains showcasing the adaptability of the proposed setting.

**Strengths:**

The problem setting is clearly motivated with a proper use case of cryo em data collection and is designed to tackle similar use case.

The problem formulation has a generalization over contextual bandits, combinatorial semi bandits, which makes it solid. Also, COAF is supported by optimality equation and design with its dependence.

The experimentation is supported by real world data to show the working validation of the motivating example. Along with it, they also show their performance on other domain with MovieLens data to showcase the wide adaptability of the setting.

Having per arm feedback within combinatorial choice helps reduces variance and seems to be realistic for their application.

**Weaknesses:**

The setting allows for switching to a new decision sets but don't signify the regime when it is optimal as supposed to exploiting.

The experimentation lacks proper baseline to compare the effectiveness of the proposed algorithm COAP.

The problem setting has IID assumption with ($X_j$ ,$A_j$ , $l_j$), however this might applications where nonstationary has to dealt with and taken into account.

**Questions:**

Since Latency aware contextual bandits seems to be the special case of contextual bandits and contextual semi bandits, If the action space and context of the arm is reduced to be similar to stochastic bandits, Does Latency aware Contextual bandits reduce to Budgeted bandits ? If so, how does the regret bound compare in this scenario ?

If a learner is allowed to request a new action set, how does this switch take latency into account, Is it already a part of the latency of the original selection action set ?

Since the work is motivated by cryo em data collection, In latency aware contextual bandit setting with COAF can it exploit all the structure in the latency observed rather than treating it as arbitrary ?

Also, For the cryo-EM, Does COAP outperforms the strong domain specific heuristic or is there any advantage of using a learned policy for cryo em data collection application ?

Also the numerical experimentation only involves a baseline comparison with the humans and Can any of the contextual bandits setting be adapted with mild relaxation to consider them for baseline evaluation ?

Often case, since cryo EM data collection involves human microscopists, drift in instrumentation or user's action changes mid run. In that case, under a IID assumption ($X_j$ ,$A_j$ , $l_j$), how does the algorithm behave ?

---

> ### Author Response · Authors · 2025-11-30
> **Response to review**
>
> We thank the reviewer for their careful consideration of our work and the valuable feedback. We recognize that there may be some misunderstanding of the core concepts, which we address below by clarifying the weaknesses and questions raised.
>
> **Exploration-Exploitation Trade-off**
> - The balance of exploration and exploitation is implicitly governed by the online estimator $\Gamma_j$. Specifically, as shown in Algorithm 1 (step 2) and Equation 3, a large estimated value for $\Gamma_j$ indicates more potential benefits to switch to new decision sets. It will lead to a more aggressive exploitation phase, where fewer arms are selected from the decision set.
>
> **Adequacy of Experimental Baselines**
> - We believe our experimental section provides a sufficient and rigorous comparison that validates the necessity and performance of our method. We utilized the MovieLens dataset for a standard comparison against several standard baselines. Crucially, the cryo-EM experiments use human operators as a benchmark, which we consider the strongest possible real-world baseline for this application domain.
>
> **Justification for Stochastic Formulation**
> - The stochastic formulation is necessary, not merely a choice. Since our model permits the selection of an arbitrary subset of arms from the decision set, the structural conditions required for a generally solvable nonstationary or adversarial setup cannot be met.
>
> - We acknowledge that standard methods like sliding windows, discounting factors, or change point detection could be incorporated in a nonstationary setup, where a limited variation budget or the number of changes is assumed. However, our paper's central contribution is the novel latency-aware decision framework itself. We maintain that addressing this unique and complex setup provides sufficient technical novelty and contribution without simultaneously introducing the additional layer of nonstationary environment modeling.
>
> **Answer to Questions**
> - We would like to stress that our formulation is a generalization of contextual bandits and contextual semi-bandits, as stated in Section 3.2. Standard approaches fail to account for this latency and will incur linear regret, as shown in Fig. 2 of the revised manuscript.. Latency-aware contextual bandits don't reduce to budgeted bandits since there is no predefined budget, and the arm filtering depends on a learned $\Gamma_j$.
>
> - The action selection of COAF (step 2 of Algorithm 1) has already taken latency into account. For example, in the cryo-EM experiment, COAF estimates the reward from a hole and evaluates whether it is worth taking time to expose it.
>
> - The latency observed in cryo-EM data collection depends on microscope conditions. COAF is designed precisely to exploit this setup, in which $\Gamma_*$ depends on the latency, as mentioned in Remark 2.
>
> - In the experimental results detailed in Section 6.2, the COAF algorithm outperforms human operators possessing several years of cryo-EM data collection experience, validating the efficacy of our model in a high-stakes, real-world setting.
>
> - We are not aware of other contextual bandit settings that can be adapted to address the action latency in cryo-EM data collection, since they normally assume unit action time. We can also provide the results of the thresholding bandit algorithm in Fig. 2(a), but we believe the human operators are stronger baselines.
>
> - The COAF algorithm enables fully automated cryo-EM data collection, operating without the intervention of human microscopists. Given the relatively slow drift rate characteristic of modern transmission electron microscopes, this environmental factor is considered negligible over the duration of a single data collection session. Therefore, the IID assumption is sufficient.

---

### Official Review · Reviewer_f5wW · 2025-11-04

**Soundness:** 2
**Presentation:** 2
**Contribution:** 2
**Rating:** 2
**Confidence:** 2

**Summary:**

The paper investigates a latency-aware contextual bandit problem, where each action (arm) incurs a random latency drawn from an unknown distribution. To capture the impact of latency on decision-making, the authors model the problem as a Markov Decision Process (MDP) and derive the corresponding Bellman optimality equation. Building upon this formulation, they propose an arm filtering algorithm that balances exploration and exploitation by accounting for both reward and latency. The proposed approach is demonstrated through experiments on the MovieLens 1M dataset and a Cryo-EM experimental setting.

**Strengths:**

The paper has the following strengths:
- The paper provides a theoretical formulation by modeling the latency-aware contextual bandit problem as an SMDP and deriving the corresponding Bellman optimality condition.
- The paper introduces a contextual online arm filtering (COAF) algorithm based on the derived Bellman condition and establishes regret bounds for both linear and general reward function settings.
- The problem is well-motivated by a real-world Cryo-EM application, and the proposed method is empirically validated on both MovieLens 1M and Cryo-EM datasets.

**Weaknesses:**

The weaknesses are described below.
- Although the paper formulates the latency-aware contextual bandit problem as an MDP, it does not clearly justify why the proposed method is preferable to existing MDP-based solutions.

- The arm filtering design and regret analysis follow relatively standard techniques, and the paper does not clearly articulate new analytical challenges introduced by latency or contextual dependencies.

- The study focuses solely on the stochastic setting, which can already be addressed by conventional MDP algorithms. Extending the formulation to adversarial or non-stationary environments would make the contribution more compelling.

- The impact of action latency on the learning rate or convergence behavior is not explicitly analyzed or reflected in the algorithmic design, despite being central to the problem motivation.

- The experimental evaluation is limited to the proposed method without comparisons against existing delayed-feedback bandits [1,2] or MDP-based algorithms, which weakens the empirical evidence supporting the algorithm’s effectiveness.


[1] Masoudian, S., Zimmert, J. and Seldin, Y., 2022. A best-of-both-worlds algorithm for bandits with delayed feedback. Advances in Neural Information Processing Systems, 35, pp.11752-11762.

[2] Lancewicki, T., Rosenberg, A. and Mansour, Y., 2022, June. Learning adversarial markov decision processes with delayed feedback. In Proceedings of the AAAI Conference on Artificial Intelligence (Vol. 36, No. 7, pp. 7281-7289).

**Questions:**

Please see the weaknesses.

---

> ### Author Response · Authors · 2025-11-29
> **Response to paper review**
>
> We appreciate the review effort, but it is clear there is a fundamental misinterpretation of our problem formulation and the scope of its technical novelty. We must firmly correct the reviewer's claims below, as they reveal a lack of appreciation for the values of our model.
>
> - Our problem is a specific, non-trivial instance of a semi-Markov decision process (SMDP) where action execution time is a critical, stochastic component that influences subsequent state transitions and decisions.
>
> - The existing solutions for standard MDPs and even many SMDP variants cannot be directly applied. Our state space is effectively unlimited and cannot be embedded naively because the state must account for the stochastic time required for ongoing actions. We are unaware of any existing solution that addresses this specific combination of unlimited state space and action-dependent time costs within this learning framework.
>
> - The technical challenge lies in learning the complex distributions governing action latency, contexts, and rewards simultaneously. Our solution successfully converts this highly complicated learning task into a more manageable problem: learning the optimal threshold function $\Gamma_*$ and the reward function.
>
> - We believe the derivation of the regret bound, which explicitly incorporates the uncertainty of action latency, is a significant technical contribution and presents a novel challenge that is not found in standard delayed-feedback or stochastic setting analyses.
>
> - The reviewer claimed, "The study focuses solely on the stochastic setting, which can already be addressed by conventional MDP algorithms." The stochastic formulation is essential and not merely a choice. Since our problem allows for the selection of an arbitrary subset of arms from the decision set, the necessary conditions for a solvable adversarial setup (e.g., standard nonstochastic bandits) are not met. The baseline policy in the nonstochastic bandit, i.e., selecting the arm with maximum total reward, will not hold in our setup. The complexity introduced by subset selection makes an adversarial framework computationally intractable or ill-defined in this context.
>
> - The impact of action latency on the regret (a metric of convergence in the bandit problem) is specifically shown in Lemma 2.
>
> - We would like to once again stress that the existing delayed-feedback bandits or MDP-based algorithms can not solve our problem, so they are compared in the experiment section. The action execution time is not equivalent to delayed-feedback. While our problem is formally related to SMDPs, the fundamental challenge of incorporating stochastic, unlimited-state-space (contexts and action latency) ensures that existing algorithms cannot solve our problem directly.  Our technical contributions provide a novel and necessary approach to this unique setting.

---

### Official Review · Reviewer_9zTZ · 2025-11-04

**Soundness:** 3
**Presentation:** 3
**Contribution:** 2
**Rating:** 4
**Confidence:** 4

**Summary:**

This paper examines a version of the contextual combinatorial bandit problem in which each action (a subset of arms) incurs a latency—a variable time cost. The goal is to maximize expected reward per unit of elapsed time rather than per round.

The authors model this as an average-reward semi-Markov decision process (SMDP) and derive a Bellman optimality equation of the form
E_{(X,A,l)}\!\left[\min_{A\in\mathcal{A}}\{l(A)\Gamma - \sum_{i\in A}\mu_i\}\right] = 0,
where \Gamma represents the long-run average reward rate.
They then propose an algorithm, COAF (Contextual Online Arm Filtering), that combines a Robbins–Monro–type stochastic approximation for estimating \( \Gamma^\* \) with UCB-style exploration for learning the arm rewards \mu_i(x).

Regret bounds of order O(T^{3/4}) are proved under both linear and general function classes, and experiments on synthetic data (MovieLens) and a cryo-electron microscopy (cryo-EM) simulation are presented.

**Strengths:**

1) The latency-aware formulation is conceptually relevant to real scientific workflows.
2) The mathematical derivations are careful and correct.
3) The paper is generally well written and easy to follow.
4) The cryo-EM example adds color and a nice application context.

**Weaknesses:**

1.	The regret bound is very likely suboptimal.
	2.	There is no lower bound or discussion of optimality.
	3.	The “latency” feature mostly amounts to a time-rescaling, I think; it is not clear why this warrants a fundamentally new theory.
	4.	The experiments lack statistical rigor—no error bars or serious baselines.
	5.	Overall novelty is modest: the algorithm is a straightforward hybrid of known tools (UCB + stochastic approximation).

**Questions:**

1.	Do you believe the T^{3/4} rate is unavoidable, or is it simply an artifact of your analysis?
	2.	When latencies are known and bounded, why can’t the setting be reduced to a contextual bandit with a random time clock?
	3.	Could one obtain a sharper \sqrt{T}-type result using a ratio or Dinkelbach-style formulation?
	4.	What exactly does “throughput” measure in the cryo-EM experiment, and how does it relate to \(\Gamma^\*\)?
	5.	Please clarify whether the cryo-EM data come from real microscope logs or a synthetic simulator.

---

> ### Author Response · Authors · 2025-11-30
> **Response to paper review**
>
> We thank the reviewer for the informative feedback. We are able to address the looseness of the regret bound. Please check Lemma 2, Theorem 3, and Theorem 4 in our revised manuscript.
>
> **Improved Regret Bounds**
>
> - Since the latency-aware contextual bandit problem generalizes the standard contextual bandit, the $\Omega(\sqrt{T})$ regret lower bound still holds for our problem.
>
> - We realize a high-probability bound for Lemma 2 is not needed. So we can remove $Q_T(\delta)$ in all regret bounds and achieve $\sqrt{T}$ type regret upper bounds. The regrets of COAF match well with their counterparts in contextual bandits, and we have made this point clear in Remark 3 and Remark 4 of the revised manuscript.
>
> **Clarification of Technical Challenge**
>
> - The reviewer claimed "The latency feature mostly amounts to a time-rescaling". In fact, the technical challenge in our problem goes far beyond a simple time-rescaling. In standard bandit problems, selecting arms from a decision set incurs a unit of time. In our model, the execution time depends on the action taken. In cryo-EM experiments, exposing more holes takes a longer time.
>
> - In our problem, the learner is required to systematically balance the potential reward and the time cost. In the cryo-EM experiment, our algorithm estimates the reward from a hole and evaluates whether it is worth taking time to expose it. The decision depends on $\Gamma_j$, which is the estimated opportunistic mean reward of switching to a new decision set.
>
> **Clarification of Experiments**
>
> - The reviewer claimed "The experiments lack statistical rigor—no error bars or serious baselines". In fact, we have provided quantile plots of regret for MovieLens experiments, which are more informative than error bars. We also added two baselines in Figure 2 of the revised manuscript, showing that standard contextual bandit algorithms can not achieve sublinear regret. Since real data is used in cryo-EM experiments, we can not run our algorithms multiple times and plot error bars.
>
> - The “throughput” in cryo-EM is formally defined as the quantity of valuable output (i.e., high-quality micrographs or collected particles) obtained per unit of time (e.g., per hour). This metric is critically dependent on the design of the reward function used in the cryo-EM data collection process. We have added Figure 3(d) in the revised manuscript for clarity.
>
> - The cryo-EM data come from real microscope logs recording the exposure time for each hole and stage movement time.
>
> **Clarification of Novelty**
>
> We strongly disagree with the assertion that the overall novelty is modest or that the algorithm is a "straightforward hybrid of known tools  (UCB + stochastic approximation)." This claim overlooks the core technical contribution achieved by our specific formulation and subsequent algorithmic conversion.
>
> - The critical difficulty of our problem lies in the fact that the learner must simultaneously estimate the complex, coupled distributions governing action latency, contexts, and rewards. Solving this directly is computationally intractable. Our primary contribution is the successful conversion of this highly complicated learning task into a more tractable problem: learning the optimal threshold function $\Gamma_*$ and the reward predictor.
>
> - We reject the use of reliance on established tools as a critique of novelty: most state-of-the-art papers in reinforcement learning and bandits build upon foundational concepts like UCB, Q-learning, or stochastic approximation. Novelty is found in how these tools are adapted and combined to solve a previously unsolved or intractable problem instance.
>
> - The novelty is further established by the scope of the problem addressed: our work provides the first learning framework for this specific latency-aware SMDP setting. As explicitly noted in Section 3.2, while the concept of "mortal bandits" (related to planning under action execution time) has existed for over 18 years, we are the first to successfully reformulate the associated planning problem into an operational online learning problem with provable regret guarantees.

---

### Author Response · Authors · 2025-12-01
**Rebuttal Summary**

We sincerely thank the reviewers for their time and effort in providing constructive and informative feedback on our submission. We have carefully addressed all comments and concerns in the attached point-by-point rebuttal and incorporated them into the rebuttal revision.

We trust that the reviewers and Area Chair will consider the following significant changes during the final evaluation of our paper:

- **Resolution of Analytical Tightness:** We have successfully addressed the concern regarding the looseness of our analysis, as raised by Reviewer 9zTZ. We have refined the theoretical bounds to rigorously achieve a $\tilde{O} (\sqrt{T})$ type regret bound, affirming the efficiency of our algorithm.

- **Strengthening Empirical Evidence:** To definitively demonstrate the inadequacy of existing methods for our problem setting, we have added two critical baselines, LinUCB and LinTS, to the MovieLens experiments. The results confirm that standard bandit algorithms fail to achieve sublinear regret in our latency-aware environment.

- **Addressing Misinterpretations of Novelty:** We acknowledge the fundamental misinterpretation of our problem formulation and novelty, particularly noted by Reviewer f5wW. Our rebuttal provides a detailed explanation clarifying that our approach addresses a non-trivial instance of the Semi-Markov Decision Process (SMDP) where stochastic action latency necessitates a novel algorithmic conversion, thus moving beyond the scope of conventional MDP or standard bandit solutions.

We look forward to the reviewers' re-evaluation of our revised submission.

---

### Meta-Review · Area_Chair_JuMi · 2026-01-05

**Summary:**

The primary issue with this paper is the lack of clarity regarding the proposed problem setting and its relationship to existing frameworks. Specifically, it remains unclear how the concept of "latency" affects the learner's decision-making process, both in terms of observation and action constraints. It is difficult to discern whether "latency" refers simply to feedback delays or if it imposes fundamental constraints on subsequent actions. This ambiguity has led several reviewers to question the novelty and the positioning of the work. While the authors attempted to address this in their rebuttal, they failed to provide a formal and concrete explanation of how latency is modeled, leaving the core of the problem unclarified.

**Reviewer Concerns:**

Addressed by Rebuttal:

The authors expanded the experimental evaluation by including additional data and comparisons. There is some room to marginally upgrade the assessment of the empirical results based on these revisions.

Outstanding Concerns:
1. Problem Formulation: The most critical concern, the formalization of "latency", remains unresolved. The rebuttal did not clarify how this concept integrates into the decision-making process, which is essential for evaluating the paper's contribution.
2. Technical Novelty: The authors provided counter-arguments regarding technical novelty, but the logic was not sufficiently persuasive to convince the reviewers, especially given the vague problem setting.

**Reviewer Scores:**

Given the persistent ambiguity of the core model, I anticipate that the scores would not have improved significantly even with a full discussion:

For most Reviewers: I predict that the scores would have remained unchanged. As long as the fundamental mechanism of "latency" remains poorly communicated, reviewers cannot accurately assess the technical non-triviality or the significance of the results.

Conclusion: The authors need to fundamentally improve their presentation to clearly show how "latency" constrains observations and actions within the decision-making process. Since the current manuscript fails to establish a clear and formal problem setting, it is not yet ready for publication at ICLR. I recommend a Reject.

---

### Decision · Program_Chairs · 2026-01-26

Reject